# FlowLet: Conditional 3D Brain MRI Synthesis using Wavelet Flow Matching

## Abstract

Brain Magnetic Resonance Imaging (MRI) plays a central role in studying neurological development, aging, and diseases. One key application is Brain Age Prediction (BAP), which estimates an individual's biological brain age from MRI data. Effective BAP models require large, diverse, and age-balanced datasets, whereas existing 3D MRI datasets are demographically skewed, limiting fairness and generalizability. Acquiring new data is costly and ethically constrained, motivating generative data augmentation. Current generative methods are often based on latent diffusion models, which operate in learned low dimensional latent spaces to address the memory demands of volumetric MRI data. However, these methods are typically slow at inference, may introduce artifacts due to latent compression, and are rarely conditioned on age, thereby affecting the BAP performance. In this work, we propose FlowLet, a conditional generative framework that synthesizes age-conditioned 3D MRIs by leveraging flow matching within an invertible 3D wavelet domain, helping to avoid reconstruction artifacts and reducing computational demands. Experiments show that FlowLet generates high-fidelity volumes with few sampling steps. Training BAP models with data generated by FlowLet improves performance for underrepresented age groups, and region-based analysis confirms preservation of anatomical structures.

## 1 Introduction

Brain MRI provides a non-invasive and high-resolution view of brain structure, playing a pivotal role in understanding neurological development, aging, and disease. An important application leveraging MRI data is Brain Age Prediction (BAP), which estimates an individual's biological brain age as a biomarker of cognitive decline and neurological disorders, thus supporting early diagnosis and treatment planning Cole et al. (2017); Baecker et al. (2021). Reliable BAP models require large, diverse, and age-balanced datasets that cover the full human lifespan. However, existing publicly available 3D MRI datasets often suffer from significant demographic imbalances, with overrepresentation of certain age groups (e.g., young adults) and underrepresentation of others (e.g., children and older adults) Bashyam et al. (2020), while large-scale datasets Sudlow et al. (2015) typically require paid access. These imbalances reduce the generalizability of the models and introduce clinical biases, limiting the applicability of BAP in real-world epidemiological and clinical settings Dinsdale et al. (2021). Collecting new MRI data to address these gaps is expensive, time-consuming, and raises ethical considerations related to patient privacy and exposure. Consequently, synthetic data augmentation through generative modeling has emerged as a promising strategy to enrich datasets and improve model robustness Chintapalli et al. (2024). Despite progress, critical challenges remain. For instance, Diffusion Models and autoencoding architectures struggle with the high dimensionality of volumetric MRI data, leading to inefficiencies and limited anatomical accuracy Dhariwal & Nichol (2021). To reduce computational demands, many models rely on latent space compression, which can introduce artifacts Müller-Franzes et al. (2023). Furthermore, insufficient conditioning mechanisms limit the preservation of fine-grained anatomical features related to age, essential for BAP. The scarcity of openly accessible implementations further restricts practical use. These technical limitations reduce the realism of the generated volumes and hinder their effectiveness in clinical tasks that require precise control over anatomical conditions.

In this work, we introduce *FlowLet*, a framework for generating conditional 3D brain MRIs that are not only anatomically consistent but also reflect the morphological characteristics of a given target

age. The core challenge lies in overcoming the *generative modeling trilemma* Xiao et al. (2022), where improvements in sample quality, diversity, and fast sampling are inherently competing objectives, so that enhancing one often comes at the expense of the others. FlowLet aims to mitigate this trilemma by integrating Flow Matching (FM) within an invertible wavelet domain. This approach is key: by combining the wavelet with a conditional architecture, FlowLet preserves fine anatomical details and supports diversity in age-specific synthesis, without the artifacts of learned compression, while FM allows sample generation in fewer steps, offering a faster alternative to previous methods.

Furthermore, we validate FlowLet on three neuroimaging datasets spanning over 12 multi-site sources in order to address data variability and generalizability and demonstrating that the synthetic brain volumes it generates are effective in clinically relevant applications, such as BAP, particularly benefiting underrepresented age groups. To evaluate our approach beyond standard practice in generative medical imaging, we combine global, functional, and regional assessments. While global metrics such as FID, MMD, and MS-SSIM are widely used, they are often insufficient in volumetric neuroimaging, where the predominance of background voxels can mask anatomical errors. To obtain a more informative picture of generative performance, we therefore include: (i) a downstream BAP task, which assesses functional utility using a well-established clinical biomarker, and (ii) region-based anatomical metrics that quantify structural accuracy across 95 brain regions. This combination provides a more reliable evaluation than relying solely on global similarity scores and helps reveal cases where strong global metrics fail to reflect local anatomical fidelity. To facilitate adoption and reproducibility, we provide the open-source implementation. Our main contributions are:

1. We introduce *FlowLet*, a FM model leveraging wavelets for anatomically accurate, multi-scale synthesis;

2. We propose an *Age-Conditioned Architecture*, with feature-wise and spatial conditioning for fine-grained control of age effects;

3. We assess FlowLet beyond standard metrics by integrating a *region-based evaluation protocol*, to capture anatomical coherence and demonstrate its utility in BAP.

## 2 RELATED WORK

**Augmentation** Despite an increase in general-purpose data availability, 3D neuroimaging remains constrained by small cohorts and high-dimensional voxel data, limiting population diversity and statistical reliability Button et al. (2013). Deep learning models, which require large and diverse datasets to generalize well, are particularly affected. Data augmentation is thus essential for improving model robustness by artificially expanding training sets. Traditional methods such as adding noise, cropping, flipping or elastic transform, preserve labels but limit variability and, more importantly, they raise the risk of introducing distorted anatomical structures or amplify biases Wu & Suk (2017); Shorten & Khoshgoftaar (2019). These risks are especially pronounced in medical imaging, where anatomical fidelity is critical and validation often relies on expert assessment. Moreover, class imbalance remains a persistent challenge: oversampling strategies, though common, scale poorly in high-dimensional spaces and may fail to capture minority class variation Nguyen et al. (2011); Blagus & Lusa (2013). These limitations require specialised methods, i.e., synthetic data generation.

**Generative Models for 3D Synthesis** Early models like GANs generate sharp samples but often suffer from instability and mode collapse Salimans et al. (2016), while VAEs enable efficient sampling at the cost of blurry reconstructions Kingma & Welling (2014). More recently, Denoising Diffusion Models Ho et al. (2020) have achieved state-of-the-art results across all domains, including medical imaging Wyatt et al. (2022); Pinaya et al. (2022); Wu et al. (2023); Durrer et al. (2023), by learning to reverse a gradual noising process. However, their iterative nature, which numerically solves Stochastic Differential Equations, requires hundreds to thousands of steps Song et al. (2021a); Davtyan et al. (2023), posing a major bottleneck for high-resolution 3D data. Advances that operate in discrete or compressed latent spaces, such as VQ-VAEs van den Oord et al. (2017) and Latent Diffusion Models (LDMs) Rombach et al. (2022), improve sampling efficiency and global reconstruction fidelity, though often at the cost of increased artifacts or added computational overhead. The challenge of scaling LDMs to ultra-high-resolution synthesis has been addressed through specialized scheduling strategies Zhang et al. (2025a).

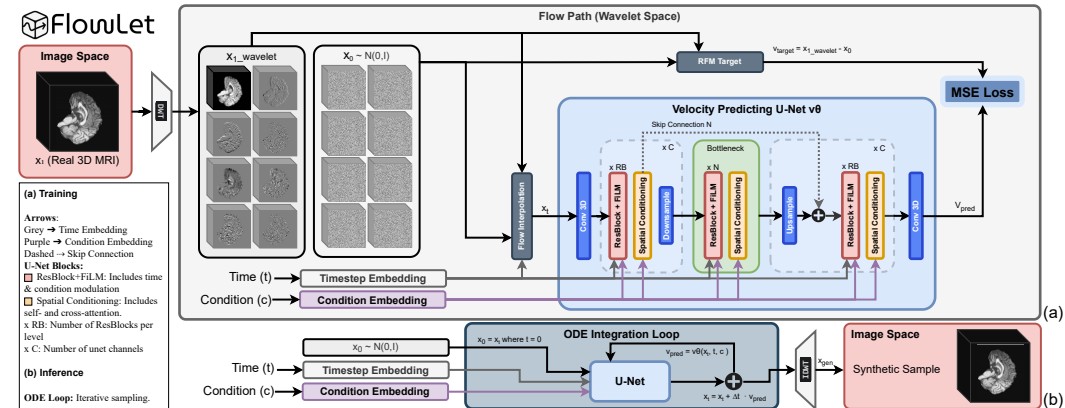

Figure 1: (a) Training in the wavelet domain decomposes the MRI into one low-frequency (LLL, black background) and seven high-frequency subbands. A conditional U-Net learns to predict velocity fields between noise and data. (b) Inference uses ODE integration followed by IDWT.

Flow Matching (FM) Lipman et al. (2023); Albergo & Vanden-Eijnden (2023); Liu et al. (2023); Neklyudov et al. (2023) mitigates this inefficiency by learning a continuous-time velocity field to transport samples from a simple prior to the data distribution via an Ordinary Differential Equation (ODE). By encouraging straighter trajectories Lee et al. (2023), FM reduces the number of required inference steps, making it convenient for fast, high-resolution volumetric synthesis. Recent efforts have focused on enhancing the conditional control and efficiency of FM Gagneux et al. (2025b)

**Wavelet Diffusion Models (WDMs)** WDMs offer a learning-free alternative to LDMs for dimensionality reduction in Diffusion Models Phung et al. (2023); Friedrich et al. (2024); Zhang et al. (2025b). WDMs use a fixed wavelet transform to decompose images into frequency components, applying diffusion directly within this wavelet domain. By removing the computational overhead of learned compression, WDMs significantly reduce memory requirements while retaining the advantages of working in a compressed, multi-resolution representation. However, the application of FM in the wavelet domain remains limited. A conceptually similar approach, combining wavelets with Normalizing Flows for high-resolution 2D image synthesis, has been previously explored in Yu et al. (2020)).

## 3 PROPOSED METHOD

FlowLet integrates FM with an invertible 3D Haar wavelet transform for efficient, anatomically coherent synthesis. A conditional network predicts velocity fields directly in the wavelet domain across multiple frequency subbands, enabling fast and accurate generation of age-specific brain volumes.

**Preliminaries** To achieve tractable learning while maintaining anatomical fidelity, the orthonormal Haar Discrete Wavelet Transform (DWT) Mallat (1999) is adopted. This nearly lossless, invertible transform decomposes 3D volumes into frequency components, reducing dimensionality while preserving structural detail Bullmore et al. (2004).

Given a 3D volume $x \in \mathbb{R}^{D \times H \times W}$, where $D$, $H$, and $W$ denote the depth (sagittal slices), height (coronal), and width (axial), respectively, the Haar DWT applies 1D low-pass ($l = \frac{1}{\sqrt{2}}[1 \ 1]$) and high-pass ($h = \frac{1}{\sqrt{2}}[-1 \ 1]$) filters sequentially along each axis. This produces 8 frequency subbands, denoted as LLL, LLH, LHL, LHH, HLL, HLH, HHL, HHH, where each letter indicates the type of filter (Low or High) applied along the corresponding axis. The resulting subbands are concatenated into a tensor: $x_w = \mathcal{W}(x) \in \mathbb{R}^{8 \times \frac{D}{2} \times \frac{H}{2} \times \frac{W}{2}}$, where $\mathcal{W}$ is the forward DWT. The transform is perfectly invertible via the inverse DWT, such that $\mathcal{W}^{-1}(x_w) = x$. Due to the orthonormality of the Haar basis, ensured by Parseval's theorem for orthonormal wavelets Mallat (1989); Daubechies (1992), energy preservation and perfect reconstruction are guaranteed across subbands.

**FlowLet Framework**   FlowLet builds upon the wavelet representation by explicitly modeling generation in the multi-scale frequency domain. As shown in Figure 1 (left), the transformed 3D MRI volume is factorized into a low-frequency approximation subband (LLL) and seven high-frequency detail subbands (LLH, LHL, ..., HHH). The LLL subband captures the dominant anatomical structure, while the remaining components isolate fine-grained, spatially localized details. As the signal energy is predominantly concentrated in the LLL subband and total energy is preserved across subbands by Parseval's theorem, interpolation trajectories in this coarse subspace are inherently smoother, supporting stable learning dynamics and reconstruction Lipman et al. (2023). At the same time, neural networks can specialize across subbands, enhancing stability and convergence in generative frameworks Ho et al. (2020); Vahdat et al. (2021). This frequency-based factorization also reduces global variability and overfitting by promoting spatial locality Mallat (1999), while significantly decreasing memory usage (8x reduction) compared to operating on full-resolution voxel volumes by drastically reducing the size of the feature maps processed by FlowLet.

## 3.1 Flow Matching Formulations

The framework supports multiple flow matching formulations to enable flexible and adaptable modeling, all operating in the wavelet domain, where samples $x_t$, noise $x_0$, data $x_1$, and velocity fields $v$ (the instantaneous time-derivative $\partial_t x_t$ along the path) belong to $\mathbb{R}^{8 \times \frac{D}{2} \times \frac{H}{2} \times \frac{W}{2}}$. The data sample $x_1$ is the wavelet transform $\mathcal{W}(x_1^{\text{voxel}})$ of an original sample $x_1^{\text{voxel}} \sim p_{\text{data}}$, and noise $x_0$ is standard Gaussian in wavelet space. FlowLet supports a modular family of FM strategies that define different continuous-time interpolation paths $x_t$ from noise to data in the wavelet domain. Each variant specifies a target velocity field $v_{\text{target}}(x_t, t)$. The model learns a parameterized velocity network $v_\theta(x_t, t, c)$, conditioned on signal $c$, by minimizing the MSE loss:

$$\mathcal{L}_{\text{FM}} = \mathbb{E}_{x_t, t, v_{\text{target}}} \left[ \| v_\theta(x_t, t, c) - v_{\text{target}}(x_t, t) \|^2 \right]. \tag{1}$$

The learned velocity field governs a deterministic ODE on $t \in [0, 1]$, solved via Euler integration.

The selected formulations, chosen for their foundational role in the FM literature and their progressively increasing trajectory curvature, are ordered to reflect differences in the geometry of the interpolation path, an aspect known to influence the expressiveness and stability of the learned velocity field Lee et al. (2023). This setup enables a systematic evaluation of how curvature impacts training stability and synthesis quality. An overview of the implemented FM variants is provided.

**Rectified Flow Matching (RFM)**   RFM Liu et al. (2023) performs linear interpolation between a Gaussian noise sample and a data sample in the wavelet domain:

$$x_t = (1 - t)x_0 + t x_1, \ v_{\text{target}} = x_1 - x_0. \tag{2}$$

The target velocity field is constant along the straight line path, resulting in zero path curvature, which promotes stable training and yields low-variance gradient estimates.

**Conditional Flow Matching (CFM)**   CFM Lipman et al. (2023); Albergo & Vanden-Eijnden (2023) constructs a linear path between a data sample $x_1$ and a sampled noise $x_0$ in wavelet space for each training instance. The target velocity field is then defined as the instantaneous direction from $x_t$ to $x_1$, scaled by the remaining time:

$$x_t = (1 - t)x_0 + t x_1, \ v_{\text{target}} = \frac{x_1 - x_t}{1 - t + \epsilon}, \tag{3}$$

where a small $\epsilon > 0$ is added to prevent divergence as $t \to 1$. While the underlying path is a straight line, the target velocity field is explicitly dependent on the current state $x_t$ and time $t$, making it more dynamic than the constant velocity of RFM. This introduces non-zero curvature and increases sensitivity as $t$ approaches 1.

**Variance-Preserving Diffusion Matching (VP)**   Inspired by DDPM Song et al. (2021a), VP defines a nonlinear interpolation from data to noise governed by a linear variance schedule $\beta(t) = \beta_{\min} + t(\beta_{\max} - \beta_{\min})$, $t \in [0, 1]$. Signal and noise scaling coefficients are:

$$\bar{\alpha}(t) = \exp\left( -\frac{1}{2} \int_0^t \beta(s)\, ds \right), \quad \sigma(t) = \sqrt{1 - \bar{\alpha}(t)^2}. \tag{4}$$

The forward noising process generates intermediate samples $x_t$ via interpolation between a data sample $x_1$ and standard Gaussian noise $\xi \sim \mathcal{N}(0, \mathbf{I})$, while the corresponding target velocity field $v_{\text{target}}(x_t, t)$, governing the reverse-time dynamics, is defined by the gradient (score) of the marginal distribution $\nabla_{x_t} \log p_t(x_t)$:

$$x_t = \bar{\alpha}(t)\, x_1 + \sigma(t)\, \xi, \; v_{\text{target}}(x_t, t) = -\frac{1}{2}\beta(t)\, x_t - \beta(t)\, \nabla_{x_t} \log p_t(x_t). \tag{5}$$

This nonlinear velocity field leads to curved reverse trajectories characteristic of diffusion models. A small positive constant is typically added to denominators during training for numerical stability.

**Trigonometric Flow** Trigonometric Nichol & Dhariwal (2021) uses a circular interpolation on the unit half-circle in wavelet space:

$$x_t = \cos\left(\frac{\pi}{2}t\right) x_0 + \sin\left(\frac{\pi}{2}t\right) x_1, \; v_{\text{target}} = \frac{\pi}{2}\left[-\sin\left(\frac{\pi}{2}t\right) x_0 + \cos\left(\frac{\pi}{2}t\right) x_1\right]. \tag{6}$$

This formulation maintains constant norm $\|x_t\|$ and has constant curvature $\frac{\pi^2}{4}$, introducing smooth curved trajectories with stable, non-straight, velocity fields.

## 3.2 Conditional U-Net Architecture

FlowLet employs a conditional 3D U-Net, $v_\theta$, designed to predict the velocity field within the 8-channel wavelet domain, as illustrated in Figure 1 (a). The model *input* consists of interpolated wavelet coefficients $x_t \in \mathbb{R}^{8 \times \frac{D}{2} \times \frac{H}{2} \times \frac{W}{2}}$, a timestep $t$, and a condition vector $c$; the *output* is the predicted velocity field $v_{\text{pred}}$ in the same wavelet domain, which is integrated via Euler ODE and mapped back to the volume domain through the IDWT. The *timestep* $t$ is embedded through sinusoidal positional encoding Vaswani et al. (2017) followed by an MLP with SiLU activations. Each *conditioning* variable is separately projected and summed to form a unified context vector $e_{\text{cond}}$. The U-Net backbone follows encoder, bottleneck, and decoder stages with skip connections to enable hierarchical feature extraction for 3D data. The primary building blocks are residual blocks (*ResBlocks*) incorporating normalization and SiLU activations.

Conditioning is incorporated depth-wise throughout the entire network using Feature-wise Linear Modulation (*FiLM*) Perez et al. (2018), allowing adaptive adjustment of activation scale and bias at every layer. Time and condition embeddings produce scale and shift parameters applied to normalized feature maps as $h_{\text{mod}} = \text{Norm}(h) \cdot (1 + \gamma_{\text{time}} + \gamma_{\text{cond}}) + (b_{\text{time}} + b_{\text{cond}})$ where $h$ represents intermediate features at each layer within the wavelet feature space. At lower spatial resolutions, where features capture more semantic information, *Spatial Conditioning* modules inspired by transformer architectures Rombach et al. (2022) are employed. These blocks first apply self-attention to model global spatial dependencies, followed by cross-attention where spatial features query the condition embedding $e_{\text{cond}}$ via $h_{\text{attn}} = \text{Attention}(Q, K, V)$, where $Q = W_Q h$, $K = W_K e_{\text{cond}}$, and $V = W_V e_{\text{cond}}$ enabling spatially-aware conditioning that refines anatomical detail based on auxiliary information. The output layer applies normalization, activation, and a convolution to produce velocity fields matching the input format in the wavelet domain. More details in Appendix A.6.

Sampling, illustrated in Figure 1 (b), is performed by solving an ODE in the wavelet domain, starting from Gaussian noise $x_0 \sim \mathcal{N}(0, I)$. The wavelet coefficients are iteratively updated using the learned velocity field $v_{pred}$. After integration, the final MRI is reconstructed by applying the IDWT.

## 4 Experimental Setup

**Datasets** The training dataset combines three publicly available sources, focusing exclusively on cognitively normal subjects. This integration was motivated by the strong age imbalance in OpenBHB[1], which despite comprising 3,984 MRI scans, it overrepresents younger individuals (primarily aged 10–20) and sparsely includes older adults. The addition of ADNI[2] and OASIS-3[3] contributed with 769 and 1,314 scans, respectively, from individuals aged 60–91 and 42–95. All MRIs were preprocessed using a standardized pipeline: bias field correction, skull stripping, spatial and intensity normalization, and resampling to a common $91 \times 109 \times 91$ resolution to facilitate BAP.

---

[1]https://baobablab.github.io/bhb/dataset

[2]https://adni.loni.usc.edu/

[3]https://sites.wustl.edu/oasisbrains/

**Baselines** For comparison, five state-of-the-art generative models are evaluated: the unconditional Wavelet Diffusion Model (*WDM*) operating in the 3D Haar wavelet domain Friedrich et al. (2024), the unconditional Medical Diffusion (*MD*) DDPM on a VQ-GAN backbone Khader et al. (2022), the conditional MONAI Latent Diffusion Model (*MLDM*) Pinaya et al. (2022), BrainSynth (*BS*) Tudosiu et al. (2024), a VQ-VAE + Transformer-based model and *MOTFM* Yazdani et al. (2025), a recent framework based on FM. Since WDM and MOTFM are unconditional in their original public implementations, we additionally introduce age-conditioned variants (WDMa and MOTFMa) to ensure a fair and rigorous comparison on our BAP task. To achieve this, we integrate our implemented conditioning mechanism directly into the respective backbones: the scalar age is mapped into a 512-dimensional embedding and subsequently injected into the UNet using cross-attention for age guidance. While other age-conditional models exist Litrico et al.; Pombo et al. (2023); Yeganeh et al. (2025), they address the distinct task of aging a specific real subject MRI rather than synthesis from noise. They are thus not directly comparable and are excluded as baselines.

**Implementation Details** FlowLet is implemented in PyTorch, using the AdamW optimizer with cosine annealing and mixed precision. For more memory-efficient attention, `xformers`[4] is optionally available. All experiments were run on an NVIDIA A6000 GPU (48 GB VRAM); notably, while baseline models required this large memory capacity, FlowLet can be trained in 24 GB, enhancing accessibility. To compare different flow formulations (RFM, CFM, VP, and Trigonometric), all variants were implemented and trained under identical architecture, hyperparameters, and optimization settings. All external baselines were likewise trained from scratch on the same dataset, using the provided default hyperparameters except for input channels and padding to match our volumes. Complete training details and hyperparameters are provided in A.7.

## 4.1 EVALUATION METRICS

As introduced earlier, generative models must balance the trilemma of producing high-quality samples, maintaining diversity and enabling efficient sampling. The evaluation pipeline is designed to reflect and investigate this balance. This is complemented with BAP to assess clinical utility, and a region-based analysis that quantifies anatomical fidelity.

**Image Fidelity and Diversity** Image quality and distributional similarity were evaluated using the Fréchet Inception Distance (FID) Heusel et al. (2017) and Gaussian kernel based Maximum Mean Discrepancy (MMD) Gretton et al. (2012), computed on features extracted from a ResNet-50 pre-trained on medical images Chen et al. (2019), following established practices for 3D medical image evaluation Friedrich et al. (2024). Lower FID and MMD indicate closer alignment to the real data distribution. The Intra-set analysis assesses the diversity of samples, calculating the average pairwise Multi-Scale Structural Similarity Index (MS-SSIM) Wang (2003), where lower values denote higher diversity. Pairwise comparisons were performed using two-sided Wilcoxon rank-sum tests with Bonferroni correction ($\alpha = 0.05$) to confirm statistical significance.

**Region-Based Anatomical Plausibility** Global similarity metrics, while useful for assessing overall image quality, can overlook fine-grained anatomical defects critical in clinical contexts. Furthermore, in volumetric neuroimaging, standard metrics such as FID and MS-SSIM are frequently dominated by empty background voxels, leading to inflated scores even when anatomical structures differ. To complement the global analysis, a region-based evaluation was performed to produce a more detailed assessment of anatomical fidelity. This involves computing three summary metrics by averaging results across the 95 cortical and sub-cortical regions of interest (ROIs), each represented by a voxel set $V_r$, extracted by `FastSurfer`[5]. Crucially, each synthetic volume is evaluated against a different age-matched real subject, and the region-based metrics are computed independently for each anatomical region $r$ and then averaged across the full set of regions $\mathcal{R}$. This one-to-one, region-wise evaluation prevents a mode-collapsed model from appearing strong, since a single repeated anatomy cannot simultaneously match the diverse set of real subjects. As a result, high scores reflect genuine anatomical plausibility and alignment with natural inter-subject variability rather than artifacts of reduced diversity. The three region-based metrics are defined as follows:

$$\text{iMAE} = \frac{1}{|\mathcal{R}|} \sum_{r \in \mathcal{R}} \left( \frac{1}{|V_r|} \sum_{v \in V_r} |R_v - S_v| \right), \quad \text{KL} = \frac{1}{|\mathcal{R}|} \sum_{r \in \mathcal{R}} D_{\text{KL}}(P_r \| \hat{P}_r), \quad \text{DICE} = \frac{1}{|\mathcal{R}|} \sum_{r \in \mathcal{R}} \frac{2|A_r \cap B_r|}{|A_r| + |B_r|}. \quad (7)$$

---

[4]https://github.com/facebookresearch/xformers
[5]https://deep-mi.org/research/fastsurfer/

Table 1: Overall mean values for synthetic sample quality. **Bold** and underlined indicate the best and second-best models per metric. The *marks results not significantly different from FlowLet-RFM 10 steps, based on pairwise Wilcoxon rank-sum tests with Bonferroni correction ($\alpha = 0.05$). Metrics are computed over 100 random bootstrap resamples of the full generated sets. Standard deviations ($\leq 10^{-3}$) are omitted for conciseness.

(a) Ours (10 steps) vs. baselines

| | Method | Steps | FID ↓ | MMD ↓ | MS-SSIM ↓ |
|---|---|---|---|---|---|
| Ours | RFM | 10 | 0.2981 | 0.0119 | 0.9508 |
| | CFM | 10 | 0.3098 | 0.0124 | 0.9707 |
| | VP | 10 | 0.3079 | 0.0123 | 0.9706 |
| | Trigon. | 10 | **0.2854** | **0.0114** | 0.9660 |
| Baselines | WDM | 1000 | 0.3073 | 0.0123 | 0.9456 |
| | WDMa | 1000 | 0.3167 | 0.0123 | 0.9430 |
| | MD | 1000 | 0.3843 | 0.0153 | 0.9595 |
| | MLDM | 1000 | 0.3590 | 0.0144 | 0.9538 |
| | MOTFM | 10 | 0.3696 | 0.0147 | 0.9676 |
| | MOTFMa | 10 | 0.3747 | 0.0145 | 0.9528 |
| | BS | – | 0.3454 | 0.0138 | **0.9346** |

Figure 2: FID vs. steps for FlowLet variants. The shaded bands indicate standard deviation.

(b) Ablations of Ours (steps + conditioning)

| Method | Steps | FID ↓ | MMD ↓ | MS-SSIM ↓ |
|---|---|---|---|---|
| RFM | 1 | 0.3334 | 0.0133 | 0.9886 |
| | 2 | 0.3232 | 0.0129 | 0.9838 |
| | 5 | 0.3130 | 0.0125 | 0.9746 |
| | 200 | 0.2978* | 0.0119* | 0.9487 |
| RFM DB4 | 10 | 0.3141 | 0.0125 | 0.9663 |
| CFM | 1 | 0.3361 | 0.0134 | 0.9899 |
| | 2 | 0.3258 | 0.0130 | 0.9858 |
| | 5 | 0.3146 | 0.0126 | 0.9771 |
| | 200 | 0.3044 | 0.0122 | 0.9508* |
| VP | 1 | 0.3341 | 0.0133 | 0.9898 |
| | 2 | 0.3234 | 0.0129 | 0.9858 |
| | 5 | 0.3132 | 0.0125 | 0.9771 |
| | 200 | 0.3004 | 0.0120 | 0.9513* |
| Trigon. | 1 | 0.3292 | 0.0131 | **0.9211** |
| | 2 | 0.2974* | 0.0119* | 0.9521 |
| | 5 | **0.2859** | **0.0114** | 0.9680 |
| | 200 | 0.3527 | 0.0141 | 0.9557 |
| Trigon. RK4 | 1 | 0.2974 | 0.0119 | 0.9485 |
| | 2 | 0.3112 | 0.0124 | 0.9576 |
| | 5 | 0.3862 | 0.0154 | 0.9604 |
| | 10 | 0.3820 | 0.0152 | 0.9579 |
| | 200 | 0.3773 | 0.0151 | 0.9518 |
| FiLM | 10 | 0.3252 | 0.0130 | 0.9861 |
| Spatial | 10 | 0.3234 | 0.0129 | 0.9846 |
| Uncond. | 10 | 0.3181 | 0.0127 | 0.9803 |

*Intensity Mean Absolute Error (iMAE)* quantifies local intensity realism by measuring the average absolute voxel-wise difference between real ($R$) and synthetic ($S$) samples within each ROI. Similarly, the overall Kullback-Leibler (KL) divergence evaluates the similarity of intensity distributions by averaging the normalized histograms of real ($P_r$) and synthetic ($\hat{P}_r$) samples across regions. Finally, the overall Dice Similarity Coefficient (DICE) assesses morphological consistency as the average spatial overlap between real ($A_r$) and synthetic ($B_r$) segmentation masks for each ROI.

These region-based anatomical fidelity metrics are intended to be interpreted alongside the diversity metric MS-SSIM. A reliable generative model should achieve both high anatomical accuracy (low iMAE, high DICE) and sufficient inter-sample variability (low MS-SSIM). Importantly, a mode-collapsed model would produce highly similar samples, yielding elevated MS-SSIM and revealing its lack of diversity, regardless of how well a single output matches a particular anatomical reference. Considering fidelity and diversity together therefore provides a more informative evaluation.

**Brain Age Prediction (BAP)** The clinical usefulness and functional fidelity of the synthetic data were evaluated by determining their impact on BAP. This task is relevant as it targets the underrepresented population (aged 44 and older), who are more susceptible to cognitive decline. Establishing a normative trajectory of healthy aging is therefore crucial, requiring the BAP task to be performed exclusively on cognitively normal (CN) subjects. Following the implementation described in De Bonis et al. (2024), 3D convolutional models were trained using synthetic data generated by each method and their performance evaluated on a real, held-out test set. BAP requires age labels to guide training and evaluation. Each generative model produced 3,000 synthetic samples, which were used to train a separate instance of a 3D BAP network. For conditional models, samples were generated across the full training age range $5.9 - 95.5$ years. To ensure a fair evaluation, ages were randomly assigned to unconditioned samples according to the age distribution of the training set. The predictor was

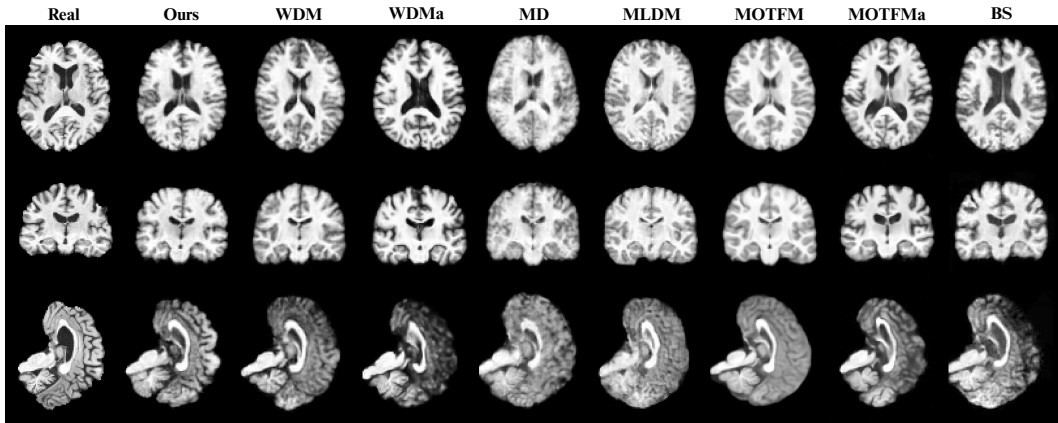

Figure 3: Visual assessment of image fidelity and realism for different 3D brain MRI synthesis models. Each column displays standard Axial, Coronal, and Sagittal views for the real reference scan (Subject of 72 years old) and specified generative method (Ours is RFM 10 steps).

evaluated on a held-out set of real subjects aged 44 and older. Prediction accuracy was measured using Absolute Error, where lower scores indicate more accurate estimation of chronological age.

## 4.2 QUALITATIVE EVALUATION

Figure 3 shows representative samples generated by each model. The FlowLet-generated volume exhibits anatomically coherent structures with well-preserved cortical folding, clearly segmented cerebellum, and identifiable subcortical regions such as the hippocampus and basal ganglia. In contrast, baselines display varying degrees of anatomical degradation, including blurring, over-smoothing in cortical areas or loss of details in posterior fossa structures. These observations suggest that FlowLet better preserves complex anatomical features while minimizing common generative artifacts.

## 4.3 QUANTITATIVE EVALUATION

**Image Fidelity and Diversity** Models were first evaluated on metrics quantifying statistical similarity between generated and real data distributions. As shown in Table 1a, all conditional FlowLet variants, evaluated with only 10 ODE steps, achieve highly competitive FID and MMD scores, surpassing the conditional MLDM, unconditional MD baselines and both MOTFM variants. Notably, Trigonometric attains the lowest FID and MMD among all tested models, suggesting that its circular interpolation path produces samples whose global feature statistics most closely match those of the real data. RFM and VP also perform on par with or better than the more computationally intensive WDM baseline. Moreover, all FlowLet variants preserve competitive MS-SSIM values, indicating that gains in fidelity are not achieved at the expense of diversity or by collapsing to a limited set of modes. The full age-stratified results and additional step ablations are provided in Appendix Table 10.

**Brain Age Prediction (BAP)** Next, the clinical relevance of the synthetic data is assessed through BAP, which provides a direct measure of the efficacy of age conditioning. The results shown in Table 2a provide several critical insights. First, they underscore the necessity of explicit conditioning for age-related tasks: Unconditional (*Uncond.*) models like WDM and MD perform poorly, demonstrating their inability to generate data with sufficient age-relevant anatomical variance for this task. Second, the FlowLet framework, particularly the RFM and CFM variants, demonstrates the best performance, achieving the lowest test MAE scores. This not only surpasses all unconditional models but also shows a substantial improvement over the strong conditional MLDM baseline. Third, the comparison between MOTFM and our conditioned MOTFMa variant validates our conditioning approach independently. Applying our conditioning mechanism to MOTFMa yields a significant performance improvement over the original MOTFM. An important finding is that both RFM and CFM-augmented training resulted in a lower test error than the BAP model trained on real data alone,

Table 2: Downstream evaluations: (a) Brain Age Prediction (BAP); (b) Region-Based.

(a) BAP for the Age $\geq 44$ years group. Lower AE = better accuracy. Ours: 10-step samples.

| | Model | Train AE $\downarrow$ | Test AE $\downarrow$ |
|---|---|---|---|
| | Real Data | $1.15 \pm 1.02$ | $4.91 \pm 3.92$ |
| Ours | RFM | $1.46 \pm 0.59$ | $4.01 \pm 3.38$ |
| | CFM | $1.39 \pm 0.59$ | $4.06 \pm 3.37$ |
| | VP | $1.02 \pm 0.49$ | $4.68 \pm 3.78$ |
| | Trigon. | $1.09 \pm 0.48$ | $4.27 \pm 3.33$ |
| Ablat. | FiLM | $0.57 \pm 0.51$ | $6.40 \pm 4.70$ |
| | Spatial | $0.87 \pm 0.54$ | $5.05 \pm 3.84$ |
| | Uncond. | $0.67 \pm 0.64$ | $5.65 \pm 3.74$ |
| Baselines | WDM | $1.63 \pm 1.36$ | $6.36 \pm 5.22$ |
| | WDMa | $0.33 \pm 0.42$ | $4.93 \pm 4.09$ |
| | MD | $2.54 \pm 2.78$ | $7.62 \pm 6.40$ |
| | MLDM | $0.98 \pm 0.47$ | $5.30 \pm 3.86$ |
| | MOTFM | $2.10 \pm 2.82$ | $10.88 \pm 9.58$ |
| | MOTFMa | $0.85 \pm 0.53$ | $4.90 \pm 3.57$ |
| | BS | $0.90 \pm 0.40$ | $4.16 \pm 3.38$ |

(b) Segmentation quality: lower is better for iMAE/KLD, higher for DICE.

| | Model | iMAE $\downarrow$ | KLD $\downarrow$ | DICE $\uparrow$ |
|---|---|---|---|---|
| Ours | RFM | $37.68 \pm 10.22$ | $0.855 \pm 0.599$ | $0.420 \pm 0.169$ |
| | CFM | $42.93 \pm 11.19$ | $1.395 \pm 1.058$ | $0.424 \pm 0.171$ |
| | VP | $37.61 \pm 10.20$ | $0.865 \pm 0.615$ | $0.423 \pm 0.171$ |
| | Trigon. | $43.35 \pm 12.06$ | $1.614 \pm 1.277$ | $0.379 \pm 0.172$ |
| Baselines | WDM | $47.52 \pm 9.45$ | $1.088 \pm 0.781$ | $0.368 \pm 0.160$ |
| | WDMa | $56.43 \pm 14.44$ | $2.112 \pm 1.090$ | $0.383 \pm 0.172$ |
| | MD | $38.44 \pm 10.44$ | $0.863 \pm 0.593$ | $0.294 \pm 0.156$ |
| | MLDM | $46.93 \pm 11.62$ | $1.040 \pm 0.645$ | $0.331 \pm 0.154$ |
| | MOTFM | $41.67 \pm 11.12$ | $0.915 \pm 0.620$ | $0.409 \pm 0.163$ |
| | MOTFMa | $42.93 \pm 11.18$ | $1.394 \pm 1.058$ | $0.391 \pm 0.162$ |
| | BS | $43.90 \pm 11.53$ | $0.862 \pm 0.589$ | $0.356 \pm 0.158$ |

indicating that FlowLet's synthetic samples can effectively mitigate data imbalances and enhance the generalization of downstream models. Finally, this functional evaluation helps to resolve the ambiguity from global fidelity metrics. While Trigonometric (*Trigon.*) achieved a highly competitive FID score, its performance on the BAP task is worse than RFM and CFM. This discrepancy suggests that while its outputs may align well in a global feature space, it may lack the fine-grained anatomical fidelity that is critical for this task. This finding, which is further corroborated by the region-based analysis in the following section, highlights that global distribution metrics do not always reflect the full spectrum of anatomical plausibility required for real applications.

**Region-Based Anatomical Plausibility**   To complement the global and functional metrics, local anatomical plausibility was assessed through a region-based analysis, with ROI metrics reported in Table 2b. The results from this fine-grained analysis show superior anatomical fidelity of the FlowLet framework, particularly for the RFM and VP variants which consistently achieve the lowest (best) iMAE and KLD scores, indicating superior local intensity and structural realism within segmented anatomical regions. They also show the highest (best) DICE scores, confirming stronger morphological consistency compared to all baselines. For instance, while the MD baseline performs competitively in intensity metrics, its significantly lower DICE score (0.294) reveals a deficiency in capturing accurate structural shapes. These findings align with the BAP task: despite a good FID, Trigonometric underperforms on regional metrics, confirming that global similarity does not imply local anatomical realism. RFM strong regional fidelity explains its superior downstream performance, highlighting the need for fine-grained and task-based evaluations in medical image synthesis.

## 4.4   ABLATION STUDIES

Ablations were conducted on: wavelet basis, inference steps, and conditioning mechanism. Full results with age-stratified metrics and statistical validation are provided in the Appendix.

**Wavelet Selection**   The choice of an appropriate wavelet basis is critical. The evaluation comprised several families, namely Haar, Daubechies (db4), Symlets (sym4), Coiflets (coif2), and Biorthogonal (bior3.3), by measuring round-trip reconstruction error (DWT followed by IDWT) on the complete training set. Haar achieved the lowest reconstruction error, with a mean MAE of $6.08 \times 10^{-8}$, despite its simplicity. The errors from the other wavelets were consistently larger, showing a degradation of at least an order of magnitude, and tended to be more spatially diffuse compared to the localized errors observed with Haar. To assess whether smoother wavelets might improve generative performance, we conducted an additional ablation in which the FlowLet-RFM model was trained using

the db4 basis. Despite the smoother filters of db4, its generative quality degraded relative to the Haar baseline: as reported in Table 1b, db4 produced a higher FID (0.3160 vs. 0.2981). Complementary evaluations in the Appendix indicate a similar trend, with higher BAP error (Table 6) and slightly reduced regional anatomical fidelity (DICE 0.4265 vs. 0.420; Table 7). The complete set of qualitative metrics for this experiment is provided in Appendix Table 11. Given its superior reconstruction fidelity and computational efficiency, attributable to its simple filters, Haar was adopted as default.

**Step Count and Efficiency**   The trade-off between the number of ODE steps and generation quality has been analyzed. As shown in Figure 2, FID generally improves with more steps, but obtains marginal gains beyond 10 steps at the cost of inference time, with Trigonometric exhibiting instability at higher step counts. The trade-off is also reflected in the sampling times: FlowLet requires approximately 0.16s for 1 step, 0.31s for 2 steps, 0.78s for 5 steps, and 1.57s for 10 steps, scaling up to 51 seconds for 200 steps. MOTFM variants achieve similar efficiency, requiring 2s for 10 steps. In contrast, baseline methods like MLDM and MD take about 12s for 1000 steps, while WDM is considerably slower, requiring around 70s for the same number of steps. Performance on the BAP task confirms that 10 steps offer an optimal balance between computational cost and functional accuracy.

**Effect of Conditioning Mechanisms**   FlowLet modulates age through two mechanisms: FiLM and Spatial Conditioning. Removing either mechanism degrades FID and MMD (Table 1b); the unconditional model still achieves competitive global scores, possibly due to global metrics being influenced by dominant patterns in the age distribution, rather than age-specific anatomical variation. Downstream evaluation exposes these differences. The full model achieves a test MAE of 4.01 years, outperforming all ablated variants. Using only Spatial Conditioning increases the error to 5.05, while FiLM alone further degrades it to 6.40. The unconditional model performs similarly poorly (5.65). These results confirm their combination to generate anatomically faithful, age-relevant samples.

**Solver Analysis**   To examine whether the instability observed for the Trigonometric path at higher step counts is a consequence of the Euler integrator, we conducted an ablation using a 4th-order Runge–Kutta (RK4) solver. The results indicate that the limitation arises primarily from the geometry of the learned velocity field rather than from the choice of numerical solver. As reported in Table 1b, RK4 yields a higher FID at 10 steps (0.3819) compared to Euler (0.2854), and converges to similarly degraded values at 200 steps. Complementary evaluations in the Appendix (Tables 6–7) show the same trend, with RK4 performing worse on both BAP and regional anatomical fidelity. Moreover, RK4 incurs a substantially higher computational cost, requiring four function evaluations per step and thereby increasing inference time by approximately a factor of four. The complete qualitative metrics for this experiment are provided in Appendix 11.

**Flow Matching Variants**   Across the studied flow-matching formulations, RFM and CFM consistently provide the most stable and reliable performance, offering a favorable balance of accuracy, robustness, and implementation simplicity. The Trigonometric path can achieve strong global metrics results at very low step counts, while exhibiting degradation as the number of steps increases. The VP formulation, did not deliver a clear performance benefit in our setting to offset its added complexity. These comparisons support our choice of RFM as the recommended formulation for FlowLet.

## 5  CONCLUSION

FlowLet integrates flow matching in the wavelet domain with age conditioning to efficiently generate anatomically faithful 3D brain MRIs, outperforming state-of-the-art baselines in fidelity, diversity, generation speed, and brain-age prediction. Among the tested variants, RFM offers the best trade-off between quality and efficiency. Nevertheless, limitations persist. Even with support from rigorous analysis and functional proxies (ROI and BAP), quantitative performance indicators alone cannot fully substitute for expert clinical evaluation, which remains essential for establishing diagnostic relevance. A critical step for future work. Although this work focused on age, the architecture can extend to multiple conditioning variables. Future work will include disease status or cognitive scores, applying FlowLet to other 3D imaging modalities, and incorporating structured expert assessment, along with ongoing bias and generalizability audits. These developments aim to advance scalable clinically meaningful MRI synthesis and support access to high-quality neuroimaging tools.

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

# A   APPENDIX

**Overview**   This section complements the main text by providing technical derivations, implementation details, and extended experimental analysis. It is intended to support transparency, reproducibility, and deeper understanding of the FlowLet framework. Specifically, it includes:

- **Flow Matching Implementation:** Direct code-level implementation details for all flow matching variants used in FlowLet. A formal derivation is included for the conditional velocity field in the Variance-Preserving formulation;

- **Data and Preprocessing:** Dataset composition, preprocessing steps, and age-normalization strategies used to construct the training and evaluation protocols;

- **Conditioning Mechanisms:** Implementation details about the dual-mechanism conditioning strategy and embeddings;

- **Training and Architecture:** Complete training configuration, model architecture details, and computational resource requirements, to facilitate reproducibility;

- **Qualitative Synthesis:** Step-wise visualizations of generated brain MRIs under different flow matching formulations, illustrating the evolution of anatomical features with varying ODE step counts;

- **Age-Stratified Evaluation:** Additional quantitative metrics computed across discrete age bins, highlighting performance variation across age groups;

- **Step-wise Ablation of Downstream Task Evaluation:** Analysis of brain age prediction performance when training with FlowLet-generated samples across varying step counts;

- **Anatomical Plausibility Evaluation:** A detailed discussion of region-wise evaluation results, including intensity realism, distributional alignment, and morphological consistency, is provided, supplemented with segmentation comparisons between synthetic and real samples;

- **Wavelet Basis Analysis:** Comparison of five wavelet families based on voxel-wise reconstruction accuracy and visual artifacts, justifying the selection of the Haar basis.

## A.1   FLOW MATCHING IMPLEMENTATIONS

This subsection clarifies the implementation of the flow matching formulations. In all cases, the training objective is to minimize the MSE loss between a predicted velocity field $v_\theta$ and a target velocity field $v_{\text{target}}$. For each training step, a time value $t$ is sampled uniformly from $[0, 1]$. The data sample $x_1$ corresponds to the variable `x1_wavelet` in the code, and the noise sample $x_0 \sim \mathcal{N}(0, I)$ is represented by variables named `x0_wavelet`.

**Rectified Flow Matching (RFM)**   The implementation directly translates the linear interpolation path and constant velocity from the paper. The path $x_t = (1-t)x_0 + tx_1$ is computed as `xt = (1 - t_broadcast) * x0_wavelet + t_broadcast * x1_wavelet`. The target velocity $v_{\text{target}} = x_1 - x_0$ corresponds to the variable `v_target = x1_wavelet - x0_wavelet`.

**Conditional Flow Matching (CFM)**   The state-dependent target velocity $v_{\text{target}} = \frac{x_1 - x_t}{1 - t + \epsilon}$ is implemented as `v_target = (x1_wavelet - xt) / (1 - t_broadcast + 1e-8)`. The term $x_t$ is computed via the same linear interpolation as in RFM, and a small $\epsilon = 10^{-8}$ is used for numerical stability.

**Trigonometric Flow**   The circular interpolation path $x_t = \cos(\frac{\pi}{2}t)x_0 + \sin(\frac{\pi}{2}t)x_1$ is implemented as `xt = torch.cos(angle) * x0_wavelet + torch.sin(angle) * x1_wavelet`, where `angle` represents $\frac{\pi}{2}t$. The corresponding velocity field $v_{\text{target}}$ is computed as its time derivative: `v_target = -torch.sin(angle) * (pi/2) * x0_wavelet + torch.cos(angle) * (pi/2) * x1_wavelet`.

## A.2 VARIANCE-PRESERVING (VP) DIFFUSION MATCHING

This section presents the explanation of the VP Diffusion Matching implementation, establishing the connection between the high-level SDE formulation from the main text and the exact conditional velocity field used for training, as implemented in the codebase. The derivation summarises and follows rigorous treatments found in the literature, particularly Song et al. (2021b); Lipman et al. (2023); Tong et al. (2023); Gagneux et al. (2025a).

**Core Definitions and Time Conventions**   The VP formulation is defined over a continuous data-to-noise time interval $t \in [0, 1]$, where $t = 0$ corresponds to clean data and $t = 1$ to pure noise. The process is governed by a linear variance schedule $\beta(t)$:

$$\beta(t) = \beta_{\min} + t(\beta_{\max} - \beta_{\min}), \tag{8}$$

where our implementation uses the standard hyperparameters $\beta_{\min} = 0.1$ and $\beta_{\max} = 20.0$ Song et al. (2021a). From this schedule, we define the signal and noise scaling coefficients, $\bar{\alpha}(t)$ and $\sigma(t)$, as follows:

$$\bar{\alpha}(t) = \exp\left(-\frac{1}{2}\int_0^t \beta(s)\,ds\right), \quad \text{and} \quad \sigma(t) = \sqrt{1 - \bar{\alpha}(t)^2}. \tag{9}$$

A noised sample $x_t$ is generated from a real sample $x_1$ via the interpolation $x_t = \bar{\alpha}(t)x_1 + \sigma(t)\xi$, where $\xi \sim \mathcal{N}(0, \mathbf{I})$.

Crucially, FlowLet training loop samples a time $t_{\text{flow}} \in [0, 1]$ along the generative noise-to-data path. This is mapped to the theoretical data-to-noise time via the relation $t = 1 - t_{\text{flow}}$. All formulas below use the theoretical time $t$.

**From Stochastic SDE to Deterministic ODE**   To make explicit the link between the VP diffusion model and the velocity field used in FlowLet, we first recall the full reverse-time SDE associated with the VP forward process  Song et al. (2021b):

$$dx_t = \left[-\tfrac{1}{2}\beta(t)x_t - \beta(t)\,\nabla_{x_t}\log p_t(x_t)\right]dt + \sqrt{\beta(t)}\,d\bar{W}_t. \tag{10}$$

For a generic diffusion SDE of the form $dx_t = f(x_t, t)\,dt + g(t)\,dW_t$, the corresponding Probability Flow ODE that evolves the same marginals $p_t$ is  Song et al. (2021a):

$$\frac{dx_t}{dt} = f(x_t, t) - \tfrac{1}{2}\,g(t)g(t)^\top\,\nabla_{x_t}\log p_t(x_t). \tag{11}$$

Applying this identity to the VP reverse SDE, with $g(t) = \sqrt{\beta(t)}I$, yields the deterministic VP flow:

$$v_{\text{ODE}}(x_t, t) = -\frac{1}{2}\beta(t)x_t - \frac{1}{2}\beta(t)\nabla_{x_t}\log p_t(x_t). \tag{12}$$

which explains the factor $1/2$ multiplying the score term. This is the velocity field that FlowLet is trained to approximate.

**Deriving the computable target velocity**   Directly using Eq. 12 for training is intractable because the true score, $\nabla_{x_t}\log p_t(x_t)$, is unknown. In the conditional flow matching framework, we circumvent this by computing the velocity conditioned on the target data sample $x_1$. This is achieved by substituting the intractable score with the analytical conditional score, $\nabla_{x_t}\log p_t(x_t|x_1)$, whose closed form is given by Tweedie's formula Efron (2011):

$$\nabla_{x_t}\log p_t(x_t|x_1) = -\frac{x_t - \bar{\alpha}(t)x_1}{\sigma(t)^2}. \tag{13}$$

Substituting this into our ODE velocity field (Eq. 12) gives the final, computable target velocity that our network learns to predict. This connection is rigorously established in recent flow matching literature Lipman et al. (2023); Tong et al. (2023); Gagneux et al. (2025a):

$$\begin{aligned}
v_{\text{target}}(x_t, t \mid x_1) &= -\frac{1}{2}\beta(t)x_t - \frac{1}{2}\beta(t)\left(-\frac{x_t - \bar{\alpha}(t)x_1}{\sigma(t)^2}\right) \\
&= -\frac{1}{2}\beta(t)\left(x_t - \frac{x_t - \bar{\alpha}(t)x_1}{1 - \bar{\alpha}(t)^2}\right) \\
&= -\frac{1}{2}\beta(t)\frac{\bar{\alpha}(t)^2 x_t - \bar{\alpha}(t)x_1}{1 - \bar{\alpha}(t)^2}.
\end{aligned} \tag{14}$$

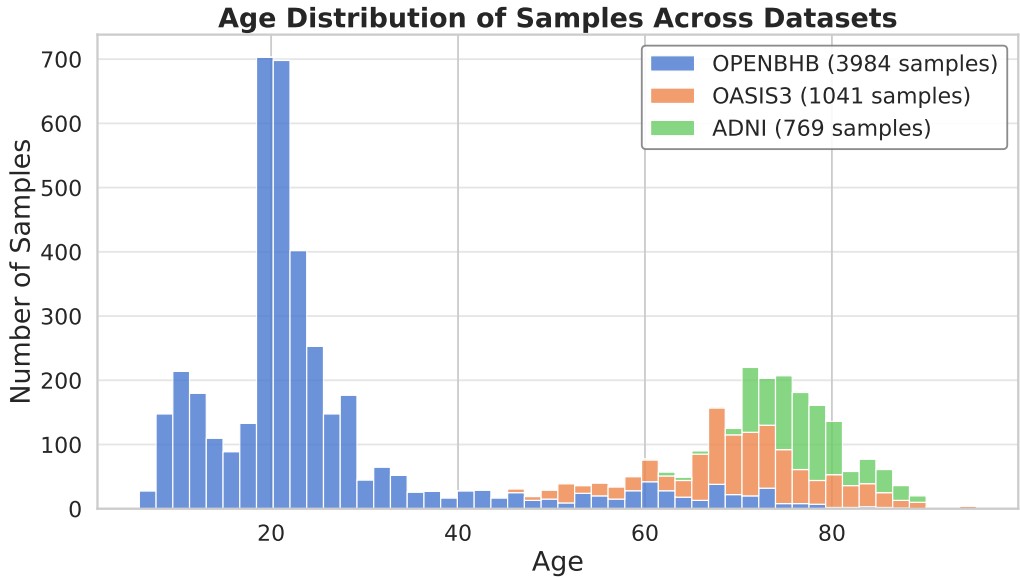

Figure 4: Combined age distribution of the training dataset after integration of OpenBHB, ADNI, and OASIS-3.

**Equivalence to the code implementation**   The target velocity in Eq. 14 is mathematically equivalent to our implementation, which adopts a numerically stable reformulation of the same expression. We recall that $\bar{\alpha}(t) = e^{-\frac{1}{2}T(t)}$ and $\bar{\alpha}(t)^2 = e^{-T(t)}$. Additionally, in the generative training loop, the time variable is reversed as $t_{\text{flow}} = 1 - t$. Finally, Eq. 14 takes the following form, as implemented in FlowLet.

$$v_{\text{target}}(x_t, t \mid x_1) = -\frac{1}{2}\beta(1-t)\frac{e^{-T(1-t)}x_t - e^{-\frac{1}{2}T(1-t)}x_1}{1 - e^{-T(1-t)}}. \tag{15}$$

All the implemented formulations are available in `/flowlet/models/flow_matching.py`.

A.3   DATASET COMPOSITION AND PREPROCESSING

The training set was constructed by integrating T1-weighted MRI scans from three large-scale, publicly available datasets: OpenBHB[6], ADNI[7], and OASIS-3[8]. All scans were filtered to include only cognitively normal individuals and were selected to ensure non-overlapping cohorts. The inclusion of ADNI and OASIS-3 supplements OpenBHB by improving age coverage and mitigating demographic imbalance, particularly within the $60 - 95$ age range. This expanded dataset supports the development and evaluation of the Brain Age Prediction (BAP) model. Below are reported relevant details for each dataset included in the experiments.

**OpenBHB**   The Open Big Healthy Brains (OpenBHB) aggregates 3,984 T1-weighted MRIs from healthy subjects across 10 publicly available datasets (e.g., IXI, ABIDE I/II, GSP, CoRR) spanning 62 imaging sites in North America, Europe, and Asia Dufumier et al. (2022). The data covers a wide age range but is heavily skewed toward young adults and adolescents. The mean age is $24.92\pm14.29$ years, with the majority of samples falling around 20 years. OpenBHB provides a predefined split into training and validation subsets, created via stratified sampling based on age, sex, and site. In the experiments, the validation subset was used exclusively for testing.

**ADNI**   The Alzheimer's Disease Neuroimaging Initiative (ADNI) is a longitudinal multi-center study aimed at tracking cognitive decline via neuroimaging biomarkers. From ADNI-1, ADNI-2,

---

[6]https://baobablab.github.io/bhb/dataset
[7]https://adni.loni.usc.edu/
[8]https://sites.wustl.edu/oasisbrains/

and ADNI-3, we selected 769 T1-weighted scans from cognitively normal individuals aged between 60-91 years (mean: 76.97±4.99). Only one scan per subject was retained to avoid repeated measures, and individuals with any cognitive impairment were excluded.

**OASIS-3** The Open Access Series of Imaging Studies (OASIS-3) is a longitudinal neuroimaging dataset that includes clinical and cognitive assessments across the adult lifespan. From this dataset, 1,314 T1-weighted MRI scans were initially identified. After filtering for cognitively normal individuals, 1,041 scans were retained, covering subjects aged between 42–95 years (mean: $71.10 \pm 8.93$).

## A.4 PREPROCESSING PIPELINE

All T1-weighted MRI volumes were processed using a standardized pipeline to ensure consistency across datasets and eliminate confounding artifacts. In particular, skull-stripping (e.g., extracting the brain from surrounding non-brain structures) plays a critical role in preventing the model from exploiting non-brain features (e.g., neck, scalp, or skull) that may correlate with age but are irrelevant to brain morphology. Such artifacts can lead to a *Clever Hans* effect Wallis & Buvat (2022), where predictions rely on spurious details rather than genuine anatomical variation. By isolating brain tissue, the model is forced to focus on age-relevant neuroanatomical patterns only. The following preprocessing steps were applied uniformly using tools from ANTs Tustison et al. (2021) and FSL:

- **Bias Field Correction:** Applied N4ITK correction Tustison et al. (2010) using ANTs[9] to remove smooth, low-frequency intensity variations that commonly affect MRI scans.
- **Spatial Normalization:** Affine registration to the MNI152 template Fonov et al. (2009) using FSL[10] FLIRT Jenkinson et al. (2002).
- **Skull Stripping:** Brain extraction performed using FSL's BET Smith (2002) on the registered images.
- **Resampling:** Volumes were resampled to an isotropic resolution of $91 \times 109 \times 91$ using ANTs. This resolution was selected for compatibility with the downstream BAP model.
- **Intensity Normalization:** Z-score normalization (zero mean, unit variance) of voxel intensities in the final volume.

Scans that failed automated preprocessing were excluded from the final dataset. A complete list of the retained samples, along with associated metadata, is provided in the codebase. No additional manual curation was performed. The full implementation is provided in the `MRI_preprocessing` folder of the codebase.

## A.5 AGE DISTRIBUTION

Figure 4 shows the age distribution of subjects in the final training set, stratified by dataset. OpenBHB contributes the majority of samples, with a pronounced peak in children and early adulthood (approximately 10–30 years), resulting in a highly imbalanced distribution. The inclusion of OASIS-3 and ADNI mitigates this skew by substantially enriching the representation of older adults (ages 60–90), leading to a more balanced coverage across the adult lifespan. The broader age coverage is necessary for training models that generalize across brain aging.

The test sets used for generative evaluation were drawn from a 20% split of the training distribution and therefore follow a similar age profile. In contrast, the BAP evaluation was conducted on the separate, independent OpenBHB test set, which follows the same distribution as the original OpenBHB cohort.

## A.6 CONDITIONAL SYNTHESIS ARCHITECTURE

A central goal of this work is not merely to generate realistic 3D brain MRIs, but to synthesize them according to specific, clinically-relevant attributes. Effectively infusing a single scalar value, such as age, into a high-dimensional generative process that defines complex anatomical structures

---

[9]https://github.com/ANTsX/ANTsPy (0.5.4)
[10]https://fsl.fmrib.ox.ac.uk/ (6.0.7.13)

Table 3: Training Hyperparameters for FlowLet Models.

| Parameter | Value |
|---|---|
| *Optimizer & Scheduler* | |
| Optimizer | AdamW |
| Learning Rate | 3e-6 |
| Weight Decay | 1e-5 |
| Adam $\beta_1, \beta_2$ | 0.9, 0.999 |
| LR Scheduler | CosineAnnealingLR |
| Scheduler `eta_min` | 1e-7 |
| *Training* | |
| Epochs | 200 |
| Batch Size | 4 |
| Gradient Clipping Norm | 1.0 |
| *Model & Architecture* | |
| U-Net Base Channels | 128 |
| U-Net Channel Multipliers | (1, 2, 4, 8) |
| U-Net Attention Resolutions | (4, 8) |
| U-Net Attention Heads | 8 |
| Condition Embedding Dim | 512 |
| `xformers` Attention | Enabled |
| Mixed Precision | Enabled |

requires a dedicated conditioning strategy. Our model, employs a dual-mechanism approach that combines pervasive, depth-wise feature modulation with targeted, spatially-aware attention. This section details the conditioning pipeline.

**The unified condition embedding ($e_{\text{cond}}$)** The architecture is designed to natively support multiple conditioning variables (e.g., Age, Sex, Disease Status). In practice, the framework accepts these arguments via configuration (e.g., `--condition_vars "Age" "Sex"`). Each conditioning information is first transformed into a unified, high-dimensional embedding vector, $e_{\text{cond}}$, which serves as the single source of conditional context for the entire U-Net. The model creates separate dedicated linear embedders for each variable and subsequently combines their projected outputs via element-wise summation to generate the final $e_{cond}$ vector.

The creation of $e_{\text{cond}}$ involves three key steps:

1. **Normalization:** Raw conditioning variables are first normalized to a consistent numerical range. For continuous variables like age, this is achieved by scaling the value to the range $[0, 1]$ based on the minimum and maximum values observed across the entire training dataset;

2. **Projection:** The normalized scalar is then projected into a high-dimensional feature space. Each condition learns a rich, non-linear mapping by being processed by its own dedicated two-layer MLP with SiLU activation, transforming the 1-dimensional normalized input into a 512-dimensional vector;

3. **Fusion:** Multiple conditions are combined by element-wise summation of their projected vectors, yielding $e_{\text{cond}}$.

This unified embedding $e_{\text{cond}}$ is then injected into the U-Net architecture using the two distinct mechanisms detailed below: FiLM for depth-wise modulation and cross-attention for spatial modulation.

**Depth-wise Conditioning via FiLM** To ensure the conditioning information influences feature computation at every level of the network, we employ Feature-wise Linear Modulation (FiLM) Perez et al. (2018) within every residual block of the U-Net. FiLM applies an affine transformation to intermediate feature maps, allowing the network to dynamically adjust the scale and bias of activations on a per-instance basis. In our model, both the flow matching timestep embedding ($e_{\text{time}}$) and the condition embedding ($e_{\text{cond}}$) generate independent scale ($\gamma$) and bias ($b$) parameters. These trans-

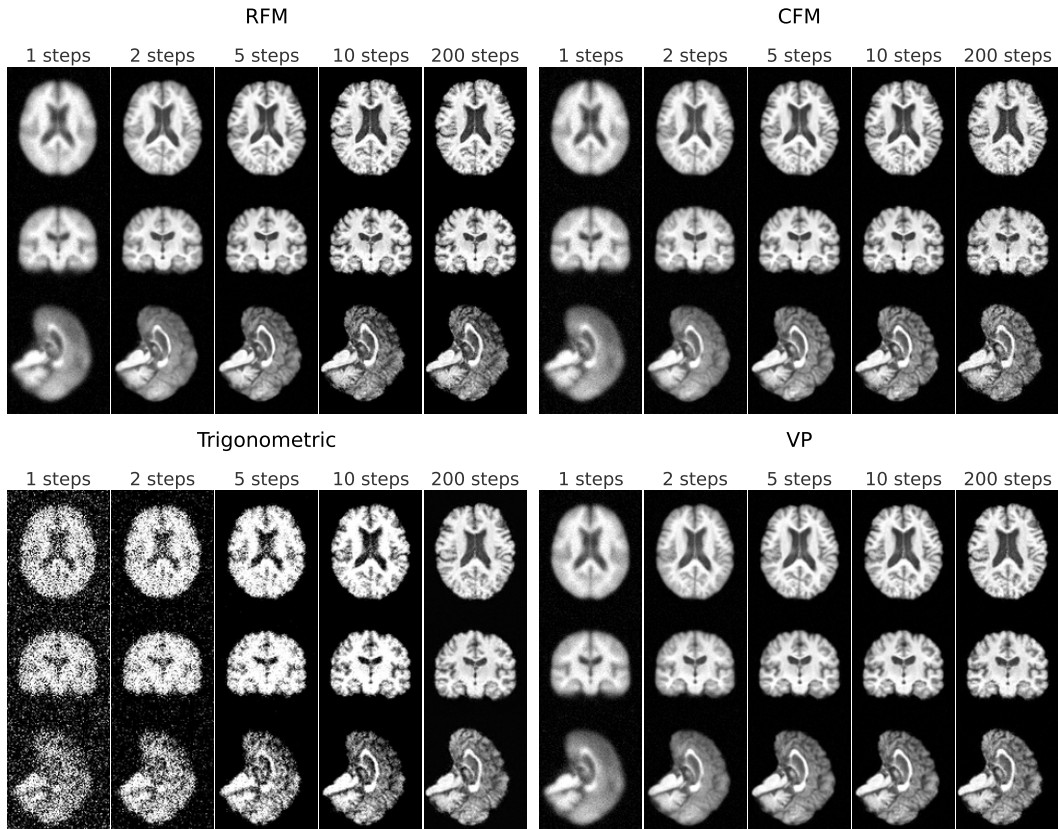

Figure 5: Qualitative comparison of FlowLet flow matching formulation across different ODE step counts. Each column shows axial, coronal, and sagittal views for a given method and step count. All samples share the same noise seed and age condition, yielding anatomically consistent outputs with similar structural compartments across flow formulations.

formations are applied sequentially to the normalized feature maps $h_{\text{norm}} = \text{Norm}(h)$ within each residual block:

$$h' = h_{\text{norm}} \cdot (1 + \gamma_{\text{time}}) + b_{\text{time}} \tag{16}$$

$$h_{\text{mod}} = h' \cdot (1 + \gamma_{\text{cond}}) + b_{\text{cond}} \tag{17}$$

The first transformation (16) allows the network to adjust features based on the current noise level in the generative trajectory. The second transformation (17) then refines these features based on the specific condition (e.g., age). By integrating FiLM into every residual block, the conditional signal is made pervasively available, influencing both low-level texture and high-level structural features as they are formed throughout the network's encoder and decoder paths.

**Spatial conditioning via Cross-Attention** While FiLM provides a powerful global modulation for each feature channel, it applies the same transformation across all spatial locations within that channel. To enable more localized, spatially-aware conditioning, we incorporate Spatial Conditioning modules inspired by transformer architectures Rombach et al. (2022) at the deeper, lower-resolution layers of the U-Net. These layers are chosen because their feature maps encode more abstract, semantic information where spatial context is critical.

Each Spatial Conditioning block first performs self-attention on the input feature map ($h$) to capture long-range spatial dependencies. This is immediately followed by a cross-attention mechanism. In this stage, the self-attended spatial features act as the *query* ($Q$), while the unified condition embedding $e_{\text{cond}}$ provides the context for both the *key* ($K$) and *value* ($V$):

$$h_{\text{attn}} = \text{Attention}(Q, K, V), \quad \text{where } Q = W_Q h, \ K = W_K e_{\text{cond}}, \text{ and } V = W_V e_{\text{cond}} \tag{18}$$

This mechanism is crucial for refining regional anatomical details, such as ventricular enlargement or cortical thinning, that are strongly correlated with age as presented in BAP and Region-Based results.

### A.7 TRAINING HYPERPARAMETERS

All models were trained using PyTorch[11] on a system running Ubuntu 24.04, equipped with a single NVIDIA A6000 GPU (48GB). Reproducibility was ensured across the entire codebase by enforcing deterministic behavior across PyTorch, NumPy, and CuDNN. The training procedure was standardized across all Flow Matching formulations and baselines to ensure fair comparison.

**Data Preparation and Loading**   The dataset was constructed from a centralized metadata file containing all MRI NIfTI file paths and associated subject metadata (e.g., age). Only subjects labeled as "Cognitively Normal" (CN) were retained. From this filtered dataset ($N = 5,794$), a deterministic 80% – 20% split was applied using a fixed random seed, resulting in 4,635 training and 1,159 validation subjects, ensuring no overlap across tasks. This split was used consistently across all training runs. The minimum and maximum age values were computed over the full dataset and used to normalize the age condition to the $[0, 1]$ range across all samples.

The preprocessing pipeline for generative modeling was applied to each 3D MRI volume and consisted of the following steps:

1. Load the NIfTI volume and normalize intensities by clipping to the 0.5th and 99.5th percentiles, then scaling to the range $[-1, 1]$.

2. Pad the volume to the required input size of $112 \times 112 \times 112$ using replication padding.

3. For training samples, apply minimal, non-invasive data augmentation using the MONAI[12] library, including random 3D rotations, intensity scaling, and Gaussian noise. These augmentations are designed to preserve anatomical structures while increasing data variance. No augmentations were applied to the validation set.

4. Apply a 3D Haar DWT, implemented using the PyWavelets[13] library, to convert the single-channel input into an 8-channel tensor representing approximation and detail subbands. This tensor is directly used as input to the U-Net.

**Training Dynamics**   The model was trained for 200 epochs using the AdamW optimizer with an initial learning rate of $3 \times 10^{-6}$, decayed via cosine annealing to a minimum of $1 \times 10^{-7}$. Automatic mixed precision (AMP) training was enabled, using a *GradScaler* to ensure numerical stability. Memory-efficient attention layers from the `xformers`[14] library were used to reduce memory consumption. Gradients were clipped to a maximum L2 norm of 1.0. The model was trained using MSE loss between the predicted and target velocity fields in the wavelet domain.

Final model selection was based on the lowest validation MSE over the complete training epochs.

The full implementation is provided in the `FlowLet_CODE` folder of the codebase.

### A.8 EFFICIENCY BENCHMARKS

To evaluate the computational efficiency of FlowLet, we report peak VRAM usage, training time, and inference throughput for all major baselines in Table 4. FlowLet requires substantially less memory during training (approximately $22\,\mathrm{GB}$ with a batch size of 4), compared to over $40\,\mathrm{GB}$ for diffusion-based baselines such as WDM and MLDM. This reduced memory footprint allows FlowLet to be trained and deployed on consumer GPUs (e.g., RTX 3090/4090). Inference is also considerably faster: FlowLet produces a full 3D sample in 1.57 seconds using 10 ODE steps, representing roughly a $45\times$ speedup over the conditional diffusion baseline WDMa (about 70 seconds for 1000 steps).

---

[11]version 2.6.0

[12]https://monai.io/ (1.4.0)

[13]https://pywavelets.readthedocs.io/ (1.8.0)

[14]https://github.com/facebookresearch/xformers (0.0.29.post3)

Table 4: Computational Resources Analysis. **BS**: Batch Size. Training times are for 200 epochs unless specified otherwise in parentheses. Inference time is measured for generating 3000 samples.

| Method | Stage / Component | BS | VRAM (GB) | Training Time | Inference |
|---|---|---|---|---|---|
| **FlowLet (Ours)** | Single Stage | 4 | 22 | 80 h | 1 h 18 min |
| MLDM | Stage 1 | 2 | 39 | 120 h | 10 h |
|  | Stage 2 | 2 | 14 | 70 h (350 ep) |  |
| MOTFM | Single Stage | 1 | 25 | 315 h | 1 h 40 min |
| MOTFMa | Single Stage | 1 | 25 | 315 h | 1 h 40 min |
| MD | Stage 1 | 4 | 16 | 74 h | 13 h 40 min |
|  | Stage 2 | 2 | 39 | 200 h |  |
|  | Stage 3 | 4 | 10 | 1 h |  |
|  | Stage 4 | 16 | 11 | 16 h 40 min |  |
| WDM | Single Stage | 1 | 40 | 96 h | 58 h 20 min |
| WDMa | Single Stage | 1 | 43 | 101 h | 58 h 20 min |
| BS | Stage 1 | 16 | 7 | 205 h (500 ep) | 4 h 18 min |
|  | Stage 2 (Extraction) | – | 4 | 1 h |  |
|  | Stage 3 | 2 | 5 | 27 h (500 ep) |  |

We further assess how FlowLet scales with increasing spatial resolution, as summarized in Table 5. FlowLet scales efficiently, requiring 42 GB of VRAM to train at $256^3$ with batch size 1. This behavior is largely due to operating in the wavelet domain. Sampling times remain stable across resolutions, increasing from roughly 1.6,s per volume at $112^3$ to 6.8,s at $256^3$ (10 ODE steps).

Table 5: Ablation study on spatial resolution, memory consumption, and speed for FlowLet model. Peak VRAM usage is reported for Batch Size (BS) 1 and 4. Inference time is measured per sample. The entry "–" indicates required VRAM exceeds 48 GB.

| Resolution | Peak VRAM (BS=1) | Peak VRAM (BS=4) | Inference Time |
|---|---|---|---|
| $112^3$ | 18 GB | 22 GB | 1.57 s |
| $128^3$ | 23 GB | 31 GB | 2.12 s |
| $256^3$ | 42 GB | – | 6.80 s |

## A.9 ADDITIONAL QUALITATIVE ASSESSMENT OF FLOW VARIANTS

To assess the generative behavior of different flow matching formulations, synthetic brain MRIs were generated using FlowLet with RFM, CFM, VP, and Trigonometric flows. For each method, volumes were sampled using 1, 2, 5, 10, and 200 ODE solver steps, with a fixed random seed and constant age condition (normalized age value $0.5 \in [0, 1]$, corresponding to 51 years). This controlled setup isolates the effect of the flow formulation and integration efficiency on the resulting images. Figure 5 displays representative axial, coronal, and sagittal mid-slices for qualitative comparison. At high step counts (e.g., 200 steps), all methods converge toward anatomically plausible structures, confirming consistency in the asymptotic regime. In contrast, notable differences arise in low-step regimes: RFM, CFM, and VP produce stable and coherent volumes with as few as 2–5 steps, while the Trigonometric flow shows instability and structural artifacts. These results illustrate the trade-off between integration curvature and sampling stability, emphasizing the advantages of lower-curvature flows for efficient and anatomically faithful synthesis.

The sample generation process with FlowLet is implemented in `scripts/generate_linear.py`.

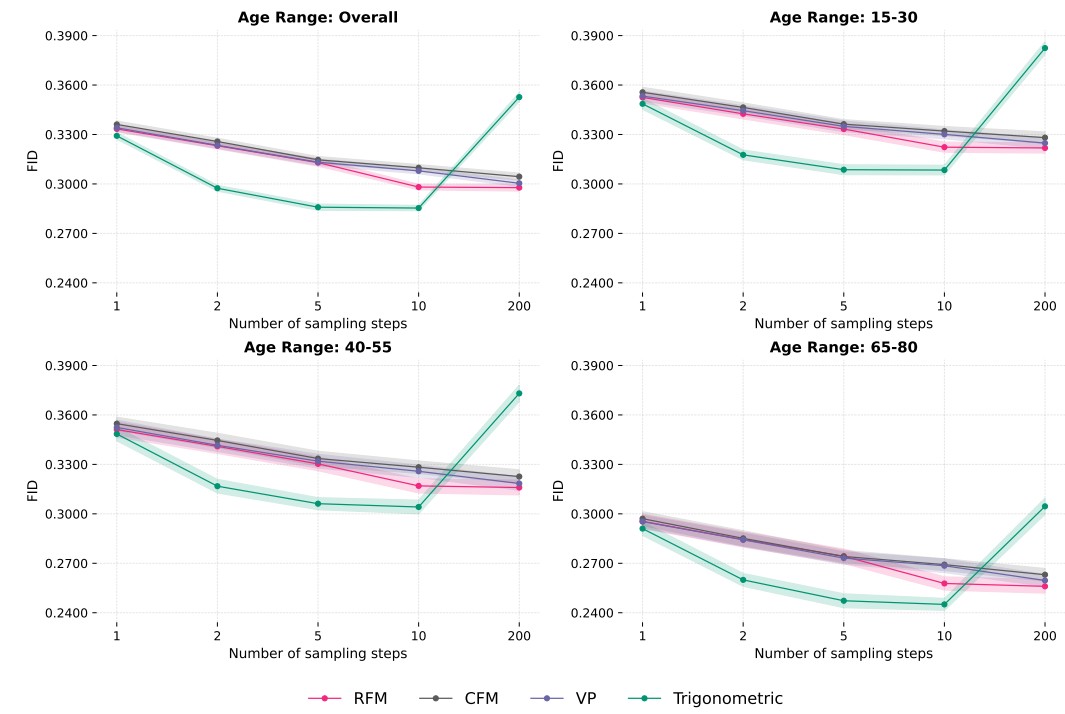

Figure 6: FID vs. number of steps for FlowLet variants calculated on overall (5.9-95 years) and by age range. The shaded bands indicate standard deviation.

## A.10 ADDITIONAL QUANTITATIVE ASSESSMENT ACROSS AGE GROUPS

To enable a finer-grained analysis of model performance, the quantitative evaluations were extended beyond the overall training age range (5.9–95.5 years) by computing metrics within discrete age subdivisions. This stratification facilitates a more detailed comparison of generative behavior across the trajectory of brain aging.

**Rationale for MS-SSIM Evaluation**     Three complementary metrics are reported for every model: FID, MMD, and MS-SSIM.

While the Multi-Scale Structural Similarity Index (MS-SSIM) is traditionally employed to assess perceptual similarity between a generated image and a reference target, it is also commonly adopted in 3D brain MRI synthesis as an *intra-set* metric, measuring structural similarity among generated samples Pinaya et al. (2022); Friedrich et al. (2024).

In this setting, MS-SSIM serves as a proxy for sample diversity:

- High intra-set MS-SSIM values (close to 1) suggest low variability among generated samples, which may indicate mode collapse.
- Low intra-set MS-SSIM values (closer to 0) can reflect excessive variation, potentially corresponding to anatomically implausible or noisy outputs.

A good generative model should produce anatomically coherent samples while preserving meaningful variability across subjects. However, MS-SSIM should not be interpreted against an absolute threshold. Rather, it is best used as a relative measure for comparison under consistent evaluation conditions.

Notably, the value of intra-set MS-SSIM can be influenced by the conditioning range used during generation. For example, generating samples within a narrow age interval is likely to result in higher similarity scores due to reduced anatomical variability, whereas broader age ranges may lead to lower MS-SSIM values as natural inter-subject differences increase.

To explore this phenomenon empirically, intra-set MS-SSIM scores were computed within three non-overlapping age bins: 15–30, 40–55, and 65–80 years. This metric was computed using all pairwise comparisons among 200 generated samples per configuration. Age conditions were uniformly sampled within each bin to ensure consistency.

The evaluation procedure is implemented in `Evaluation_FID_MMD_MSSSIM.py`.

**Benchmark Results**   Table 10 (overall, 15–30, 40–55 and 65–80 age groups) reports the complete quantitative results. Cells marked with a dash ("–") under the unconditional (*Uncond.*) baselines (RFM-Uncond., WDM, MOTFM and MD) indicate that age-stratified metrics could not be computed, as these models lack explicit age conditioning and therefore cannot be evaluated within the defined age range.

The performance trend observed in the overall evaluation is consistently reflected across all age groups. Among the evaluated age ranges, the 65–80 group achieves the best performance in terms of both FID and intra-set MS-SSIM, indicating that FlowLet produces high-fidelity and diverse samples precisely in the demographic segment where data augmentation is most needed. This group exhibits the lowest MS-SSIM values across all age bins, suggesting increased anatomical variability in later adulthood, potentially reflecting diverse neuroanatomical aging trajectories Bethlehem et al. (2022). In contrast, higher MS-SSIM scores observed in younger age groups likely reflect more homogeneous structural patterns.

To complement the results presented in the main text, Table 11 provides the complete quantitative results for the new ablation experiments, including the performance of FlowLet-RFM trained with the Daubechies-4 (db4) basis and the Trigonometric flow variant integrated using the 4th-Order Runge-Kutta (RK4) solver. The performance degradation observed in the overall metrics for the RFM-db4 variant (FID 0.3142 vs. 0.2981 for Haar) is consistent across all age bins (Table 11), confirming that the superior reconstruction fidelity of the Haar basis translates into better generalized generative performance, even in age-stratified contexts. Similarly, the RK4 integration of the Trigonometric path, while intended to improve stability, failed to yield superior results compared to the simpler Euler-integrated RFM, often showing inferior FID/MMD across multiple step counts and age groups.

These findings underscore the importance of contextualizing intra-set MS-SSIM values with respect to the demographic characteristics of the generated data. All comparisons are reported within age-matched intervals to ensure fair and meaningful evaluation of generative diversity.

**FID *vs.* Sampling Steps**   Figure 6 plots the FID as a function of ODE steps for all FlowLet variants, stratified by age group. All curves exhibit a consistent monotonic improvement with increasing step counts, in agreement with trends observed in the aggregate metrics. A performance plateau is observed between 10 and 200 steps, supporting the selection of 10 steps as an optimal balance between sampling efficiency and image fidelity across age groups. Notably, the Trigonometric flow, despite achieving competitive FID scores at low step counts, shows greater variability and degraded performance in high-step regimes.

## A.11   Additional BAP Training Details

Following the methodology described in De Bonis et al. (2024), a 3D DenseNet-121 architecture was employed for BAP, configured for regression with a linear output layer to estimate chronological age from structural T1-weighted MRI volumes. Input samples were normalized to the [0, 1] intensity range using the 5th and 95th percentile values computed from the training set to avoid test-set leakage.

The model was trained using Stochastic Gradient Descent with a cosine annealing warm restarts scheduler. The performance was evaluated on the independent test set of OpenBHB. The analysis was restricted to subjects aged 44 years and older, the demographically underrepresented group in the training distribution. The full set of hyperparameters used for training is provided in Table 8.

For the steps ablation, FlowLet with the RFM formulation was used to generate 3,000 synthetic brain MRIs per setting, varying the number of ODE solver steps (1, 2, 5, 10, 200). Conditioning was applied linearly across the full age span of the training set (5.9–95.5 years) to ensure broad age coverage. A separate BAP model was trained on each of these synthetic-augmented datasets, using

Table 6: Brain Age Prediction Performance for the Age $\geq 44$ years group on the merged dataset. Lower values indicate better prediction accuracy. Results are reported as Absolute Error (AE) and Standard Deviation.

| Model | Steps | Train AE ↓ | Test AE ↓ |
|---|---|---|---|
| Real Training Data | | $1.15 \pm 1.02$ | $4.91 \pm 3.92$ |
| Ours RFM | 1 step | $1.32 \pm 0.91$ | $4.81 \pm 4.43$ |
| | 2 steps | $1.03 \pm 0.61$ | $4.74 \pm 3.85$ |
| | 5 steps | $1.42 \pm 0.46$ | $4.23 \pm 3.52$ |
| | 10 steps | $1.46 \pm 0.59$ | $4.01 \pm 3.38$ |
| | 200 steps | $1.83 \pm 0.40$ | $4.80 \pm 3.91$ |
| Ours Trigon. RK4 | 1 step | $1.09 \pm 0.45$ | $4.26 \pm 3.40$ |
| | 2 steps | $0.33 \pm 0.40$ | $4.10 \pm 3.17$ |
| | 5 steps | $0.93 \pm 0.56$ | $4.12 \pm 3.73$ |
| | 10 steps | $0.76 \pm 0.41$ | $4.27 \pm 3.49$ |
| | 200 steps | $0.63 \pm 0.42$ | $4.43 \pm 3.83$ |
| Ours RFM db4 | 10 steps | $0.50 \pm 0.35$ | $4.61 \pm 4.33$ |

Table 7: Segmentation (ROI) quality metrics for the additional experiments. Lower values indicate better performance for iMAE and KLD; Higher values indicate better performance for DICE.

| Model | Steps | iMAE ↓ | KLD ↓ | DICE ↑ |
|---|---|---|---|---|
| Ours Trigon. RK4 | 1 step | $40.14 \pm 10.53$ | $1.186 \pm 0.676$ | $0.038 \pm 0.099$ |
| | 2 steps | $37.21 \pm 9.96$ | $0.915 \pm 0.668$ | $0.401 \pm 0.182$ |
| | 5 steps | $37.43 \pm 10.27$ | $0.868 \pm 0.628$ | $0.407 \pm 0.174$ |
| | 10 steps | $38.63 \pm 10.50$ | $0.969 \pm 0.707$ | $0.370 \pm 0.166$ |
| | 200 steps | $43.35 \pm 12.06$ | $1.614 \pm 1.277$ | $0.394 \pm 0.172$ |
| Ours RFM db4 | 10 steps | $40.48 \pm 10.85$ | $1.065 \pm 0.751$ | $0.427 \pm 0.171$ |

the same architecture and training protocol. For direct comparison of the utility and effectiveness of the synthetic samples, a reference model was trained exclusively on the full real training set.

For the Trigonometric RK4 variant, changing the solver from Euler to RK4 did not recover the instability observed at higher step counts, and the resulting BAP MAE scores (e.g., 4.27 years at 10 steps) remained inferior to the RFM baseline (4.01 years). Following the same procedure, the RFM model (10 steps) trained using the smoother Daubechies-4 (db4) basis yielded a Test MAE of 4.61 years, a notable degradation compared to the primary RFM-Haar baseline.

The implementation of the BAP training pipeline is available in the `BAP_trainer` folder of the codebase.

**BAP *vs.* Sampling Steps**  As shown in Table 6, all RFM-augmented datasets outperformed the real-only reference baseline. Notably, even the 1-step configuration led to better test performance, highlighting the clinical utility of FlowLet-generated samples even under extreme sampling constraints. The best performance was achieved at 10 steps, demonstrating an effective trade-off between sampling efficiency and anatomical fidelity. Increasing the step count to 200 did not result in further gains, suggesting that FlowLet-RFM reaches peak downstream utility with minimal inference cost.

## A.12  ADDITIONAL REGION-BASED ANATOMICAL EVALUATION

FastSurfer[15] is a deep-learning pipeline for automatic segmentation and parcellation of the brain from structural MRIs. It was leveraged to further evaluate the anatomical plausibility of the synthetic data generated by FlowLet. This analysis included baseline models (WDM, WDMa, MD, MLDM, MOTFM, MOTFMa, BS) and all FlowLet variants (RFM, CFM, VP, Trigonometric), with

---

[15]https://deep-mi.org/research/fastsurfer/ (v2.4.2)

Table 8: Training Hyperparameters for the BAP Model.

| Parameter | Value |
|---|---|
| *Optimizer & Scheduler* | |
| Optimizer | SGD |
| Initial Learning Rate | 0.01 |
| LR Scheduler | CosineAnnealingWarmRestarts |
| Scheduler `T_0` | 17 |
| Scheduler `T_mult` | 2 |
| Scheduler `eta_min` | 1e-5 |
| *Training* | |
| Epochs | 100 |
| Batch Size | 16 |
| Loss Function | L1 Loss (MAE) |
| *Model & Architecture* | |
| Model Architecture | DenseNet-121 |
| Input Channels | 1 |
| Output Classes | 1 (Age) |
| *Data Preprocessing* | |
| Normalization Method | Min-Max Scaling |
| Scaling Min Value | 5th Percentile (of training set) |
| Scaling Max Value | 95th Percentile (of training set) |

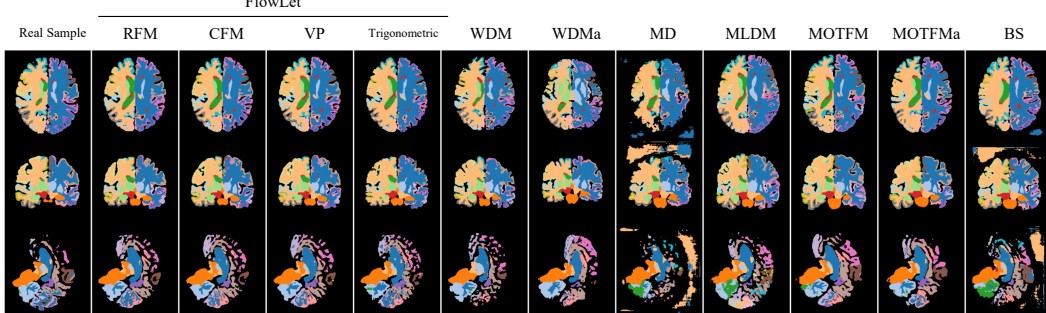

Figure 7: Qualitative comparison of FastSurfer segmentations of synthetic data from different models. The leftmost column displays the segmentation from a real 72-year-old subject, while each subsequent column shows axial, coronal, and sagittal views relative to a synthetic sample of the same age generated by a specific model. FlowLet samples are generated at 10 steps.

the latter configured for 10-step generation. Additionally, ablations on the FlowLet-RFM model using the Daubechies-4 basis and the FlowLet-Trigonometric variant integrated with the 4th-Order Runge-Kutta solver across various step counts were performed. Each model generated 500 synthetic brain samples linearly spanning the full age range of the training set using the same seed for consistency, ensuring comparable anatomical content across FlowLet variants despite differences in flow formulation. These samples were then segmented into 95 anatomical classes using FastSurfer. A reference set of 500 real samples was similarly processed for comparison.

**Region-Wise Metric Definitions** The anatomical quality of the generated samples was evaluated through a region-wise comparison between each synthetic brain volume and an age-matched real reference. For each pair, brain segmentations were obtained using FastSurfer, and three summary metrics were computed across anatomically defined regions to assess local realism and consistency:

- Overall intensity Mean Absolute Error (iMAE);

- Overall Kullback-Leibler Divergence (KL);

- Overall Dice Similarity Coefficient (DICE).

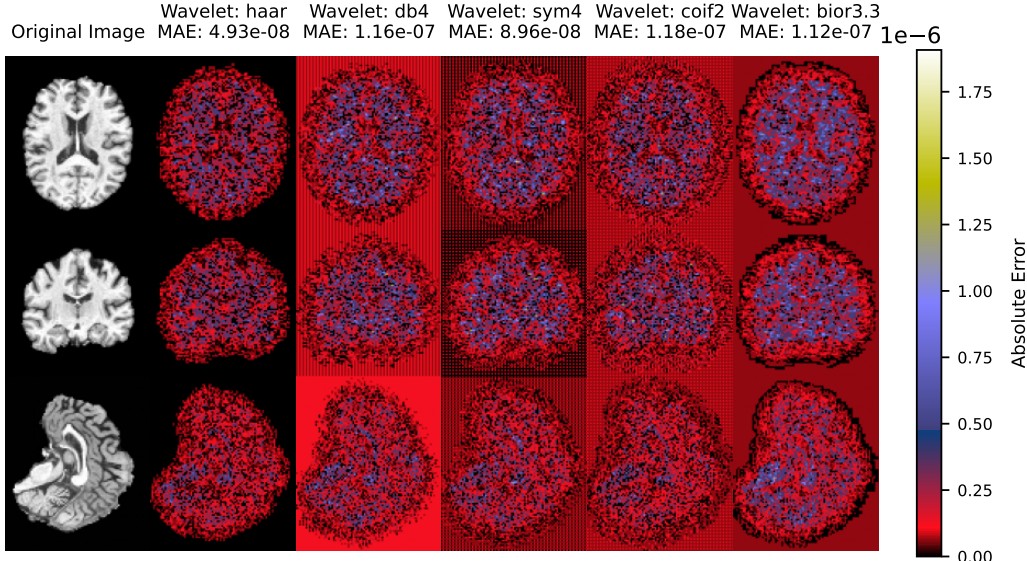

Figure 8: The leftmost column shows a real MRI volume in three views (axial, coronal, sagittal). Subsequent columns display the absolute reconstruction error for the same sample after a round-trip transform using different wavelet bases. The reported MAE values were computed over the entire set of voxel intensities in the 3D volume for this single instance. The color scale is adjusted to emphasize the narrow dynamic range of errors and facilitate visual comparison.

These metrics were designed to assess the overall anatomical quality of the generated brain volumes using field-specific criteria. Dice ranges from 0 (no overlap) to 1 (perfect overlap), while lower values of iMAE and KL indicate better intensity and distributional alignment.

In particular, the iMAE and KL require a voxel-wise comparison between each synthetic volume ($S$) and a corresponding real reference ($R$) for each ROI ($r \in \mathcal{R}$). However, due to natural anatomical variability, the ROI segmentations from synthetic and real volumes may not perfectly overlap, meaning the two versions of a given ROI may contain different voxel sets. To handle this, we define the comparison set for each ROI as the union of the voxel sets from both the synthetic and real segmentations. This ensures that all relevant voxels are included in the comparison. Each metric is then computed over this unified set and averaged across all regions ($\mathcal{R}$), providing a robust summary of the anatomical plausibility of the synthetic volume relative to the real one.

The evaluation procedure is implemented in the `ROI_Evaluation` folder of the codebase.

**Qualitative Assessment**    Figure 7 shows representative segmentations of 72-year-old samples generated by different methods and compared to a real sample of the same age. For the unconditional baselines (MOTFM, MD and WDM), the sample was selected randomly. All FlowLet-generated samples show similar segmentations. However, FlowLet-Trigonometric led to a more jagged segmentation than the others, as shown in the sagittal view. Moreover, the figure suggests that the baselines struggled in generating the occipital lobe and the cerebellum, as evidenced by segmentation outputs with more gaps and fewer anatomical details compared to the reference. Conversely, FlowLet-based segmentations show detailed regions for the occipital lobe and the cerebellum, suggesting a better representational capability of the FlowLet family for these regions.

† Five wavelet families were compared, Haar, Daubechies-4 (db4), Symlet-4 (sym4), Coiflet-2 (coif2), and Biorthogonal 3.3 (bior3.3), to assess their suitability for 3D MRI volume reconstruction. The analysis focused on filter structure, computational efficiency, reconstruction accuracy, and the presence of voxel-domain artifacts. Detailed properties, including filter definitions and theoretical foundations, are available in Mallat (1989); Daubechies (1992). Table 9 reports quantitative reconstruction errors, and Figure 8 visualizes error distributions for a representative sample.

Table 9: Mean Absolute Error (MAE) and Standard Deviation (Std) computed over the entire training set ($N = 5{,}794$) for each wavelet. The error reflects voxel-wise intensity differences between original and reconstructed volumes.

| Wavelet Type | MAE $\pm$ Std $\downarrow$ |
|---|---|
| Haar | $6.08 \times 10^{-8} \pm 1.60 \times 10^{-8}$ |
| Daubechies (db4) | $1.03 \times 10^{-7} \pm 2.30 \times 10^{-8}$ |
| Symlet (sym4) | $1.11 \times 10^{-7} \pm 2.81 \times 10^{-8}$ |
| Coiflet (coif2) | $9.98 \times 10^{-8} \pm 2.24 \times 10^{-8}$ |
| Biorthogonal (bior3.3) | $1.32 \times 10^{-7} \pm 3.70 \times 10^{-8}$ |

**Haar (db1)**    The Haar wavelet (Daubechies-1) is the simplest wavelet, defined by a step-function basis. It is discontinuous and uses a length-2 filter, providing the shortest support and highest computational efficiency. Haar achieved the lowest reconstruction error (mean MAE: $6.08 \times 10^{-8}$), with minimal boundary artifacts and numerically exact reconstruction. Although its constant basis functions can cause blockiness under compression, no such artifacts were observed under lossless reconstruction.

**Daubechies (db4)**    Daubechies-4 is an orthonormal wavelet with 4 vanishing moments and an 8-tap filter. Its smoother basis functions improve energy compaction and reduce blockiness compared to Haar. However, its longer support can introduce ringing near sharp transitions. In the experiments, db4 showed reconstruction errors roughly an order of magnitude higher than Haar (MAE $\sim 10^{-7}$), likely due to boundary effects.

**Symlets (sym4)**    Sym4 is a near-symmetric variant of db4, preserving orthonormality and using an 8-tap filter. Its symmetry helps reduce phase distortion and shift sensitivity. Like db4, its reconstruction errors remained in the $10^{-7}$ range and were primarily localized in background voids.

**Coiflets (coif2)**    Coiflet-2 uses a 12-tap, near-symmetric filter with vanishing moments in both wavelet and scaling functions. It enables smooth intensity transitions at higher computational cost and boundary sensitivity. It achieved the second-lowest reconstruction error (MAE $\sim 10^{-8}$), though some oscillatory artifacts appeared near high-contrast edges.

**Biorthogonal (bior3.3)**    Bior3.3 employs separate analysis and synthesis filters with three vanishing moments each and an 8-tap symmetric analysis filter. This biorthogonal design ensures linear phase and supports shift-invariant, edge-aligned reconstruction. The MAE was on par with other 8-tap wavelets ($\sim 10^{-7}$), with mild structured artifacts near edges, likely due to non-orthogonality.

**Qualitative Reconstruction Analysis**    Figure 8 presents the absolute voxel-wise reconstruction error for a representative MRI volume. This qualitative view highlights spatial error patterns following a single round-trip transform. Haar shows minimal, localized error within the brain, while the other bases introduce structured artifacts in background voids, reflecting longer filter supports or non-orthogonal behavior. These visual differences are consistent with the quantitative findings in Table 9.

Among all evaluated wavelets, Haar consistently achieved the best trade-off between reconstruction fidelity, computational cost, and artifact suppression, especially in low-signal regions, supporting its choice as the default basis for FlowLet.

### A.13    SIGNIFICANCE TESTING AND P-VALUE ANALYSIS

To assess the reliability of the performance differences observed in our study, we computed statistical significance relative to the reference configuration (FlowLet-RFM at 10 steps). These p-values correspond to the global quality metrics (FID, MMD, and MS-SSIM) reported in Table 10 and Table 11. In the heatmap, cells colored in red indicate results that are non-statistically significant ($p > 0.05$). Notably, the prevalence of red cells in the 200-step rows for the RFM, CFM, and VP

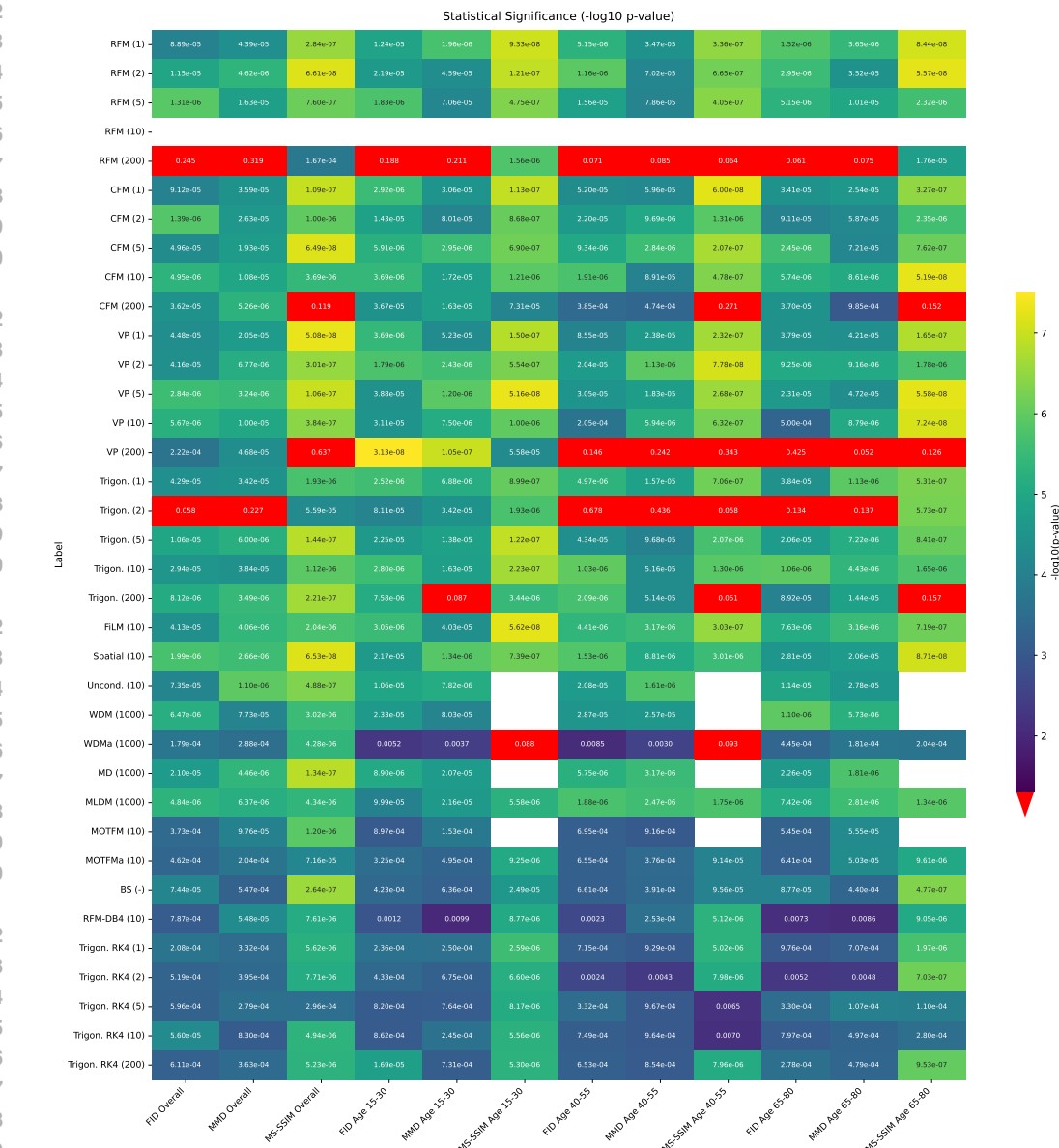

Figure 9: Statistical Significance ($-\log_{10}$ p-value) of Pairwise Comparisons against FlowLet-RFM (10 Steps). The heatmap displays the results of the Bonferroni-corrected p-values obtained from two-sided Wilcoxon rank-sum tests for pairwise comparisons between the indicated models and the FlowLet-RFM (10 steps) baseline across various fidelity, diversity, and age-stratified metrics. Cells colored red indicate a non-statistically significant difference ($p > 0.05$) from the FlowLet-RFM (10 steps) baseline.

solvers confirms that extending the inference trajectory beyond 10 steps yields statistically negligible improvements in sample quality for these methods. Conversely, the remaining colored regions (green to yellow) represent statistically significant differences, validating the distinct performance gains achieved by the Trigonometric solver and the variations observed in the baseline comparisons.

Table 10: Synthetic sample quality metrics across different age groups. Bold and underlined values are best and second-best models. The $*$ marks results not significantly different from FlowLet-RFM at 10 steps ($p > 0.05$). The metrics are computed over 100 bootstrap resamples. Values reported as "–" indicate configurations where age-specific metrics could not be computed, such as unconditional models lacking explicit age control.

| Method | Type | Steps | Overall FID↓ | Overall MMD↓ | Overall MS-SSIM↓ | Age 15–30 FID↓ | Age 15–30 MMD↓ | Age 15–30 MS-SSIM↓ |
|---|---|---|---|---|---|---|---|---|
| Ours | RFM | 1 | $0.3334 \pm 0.0018$ | $0.0133 \pm 0.0001$ | $0.9886 \pm 0.0102$ | $0.3525 \pm 0.0033$ | $0.0142 \pm 0.0001$ | $0.9981 \pm 0.0002$ |
| | | 2 | $0.3232 \pm 0.0018$ | $0.0129 \pm 0.0001$ | $0.9838 \pm 0.0112$ | $0.3425 \pm 0.0031$ | $0.0138 \pm 0.0001$ | $0.9942 \pm 0.0004$ |
| | | 5 | $0.3130 \pm 0.0022$ | $0.0125 \pm 0.0001$ | $0.9746 \pm 0.0121$ | $0.3333 \pm 0.0033$ | $0.0134 \pm 0.0001$ | $0.9863 \pm 0.0008$ |
| | | 10 | $0.2981 \pm 0.0017$ | $0.0119 \pm 0.0001$ | $0.9508 \pm 0.0195$ | $0.3223 \pm 0.0030$ | $0.0130 \pm 0.0001$ | $0.9698 \pm 0.0043$ |
| | | 200 | $0.2978^* \pm 0.0020$ | $0.0119^* \pm 0.0001$ | $0.9487 \pm 0.0206$ | $0.3218^* \pm 0.0031$ | $0.0130^* \pm 0.0001$ | $0.9687 \pm 0.0047$ |
| Ours | CFM | 1 | $0.3361 \pm 0.0021$ | $0.0134 \pm 0.0001$ | $0.9899 \pm 0.0093$ | $0.3556 \pm 0.0030$ | $0.0143 \pm 0.0001$ | $0.9978 \pm 0.0005$ |
| | | 2 | $0.3258 \pm 0.0019$ | $0.0130 \pm 0.0001$ | $0.9858 \pm 0.0104$ | $0.3464 \pm 0.0027$ | $0.0139 \pm 0.0001$ | $0.9945 \pm 0.0006$ |
| | | 5 | $0.3146 \pm 0.0020$ | $0.0126 \pm 0.0001$ | $0.9771 \pm 0.0111$ | $0.3363 \pm 0.0027$ | $0.0135 \pm 0.0001$ | $0.9870 \pm 0.0009$ |
| | | 10 | $0.3098 \pm 0.0019$ | $0.0124 \pm 0.0001$ | $0.9707 \pm 0.0117$ | $0.3321 \pm 0.0027$ | $0.0134 \pm 0.0001$ | $0.9815 \pm 0.0014$ |
| | | 200 | $0.3044 \pm 0.0021$ | $0.0122 \pm 0.0001$ | $0.9508^* \pm 0.0182$ | $0.3281 \pm 0.0034$ | $0.0132 \pm 0.0001$ | $0.9675 \pm 0.0055$ |
| Ours | VP | 1 | $0.3341 \pm 0.0021$ | $0.0133 \pm 0.0001$ | $0.9898 \pm 0.0092$ | $0.3533 \pm 0.0030$ | $0.0142 \pm 0.0001$ | $0.9979 \pm 0.0003$ |
| | | 2 | $0.3234 \pm 0.0018$ | $0.0129 \pm 0.0001$ | $0.9858 \pm 0.0101$ | $0.3445 \pm 0.0029$ | $0.0138 \pm 0.0001$ | $0.9948 \pm 0.0004$ |
| | | 5 | $0.3132 \pm 0.0019$ | $0.0125 \pm 0.0001$ | $0.9771 \pm 0.0109$ | $0.3349 \pm 0.0032$ | $0.0135 \pm 0.0001$ | $0.9871 \pm 0.0008$ |
| | | 10 | $0.3079 \pm 0.0018$ | $0.0123 \pm 0.0001$ | $0.9706 \pm 0.0115$ | $0.3301 \pm 0.0026$ | $0.0133 \pm 0.0001$ | $0.9817 \pm 0.0013$ |
| | | 200 | $0.3004 \pm 0.0019$ | $0.0120 \pm 0.0001$ | $0.9513^* \pm 0.0183$ | $0.3248 \pm 0.0032$ | $0.0131 \pm 0.0001$ | $0.9685^* \pm 0.0057$ |
| Ours | Trigon. | 1 | $0.3292 \pm 0.0020$ | $0.0131 \pm 0.0001$ | $\mathbf{0.9211} \pm 0.0084$ | $0.3486 \pm 0.0033$ | $0.0140 \pm 0.0001$ | $\mathbf{0.9292} \pm 0.0004$ |
| | | 2 | $0.2974^* \pm 0.0017$ | $0.0119^* \pm 0.0001$ | $0.9521 \pm 0.0112$ | $0.3176 \pm 0.0029$ | $0.0128 \pm 0.0001$ | $0.9624 \pm 0.0004$ |
| | | 5 | $0.2859 \pm 0.0019$ | $\mathbf{0.0114} \pm 0.0001$ | $0.9680 \pm 0.0130$ | $0.3086 \pm 0.0030$ | $\mathbf{0.0124} \pm 0.0001$ | $0.9799 \pm 0.0008$ |
| | | 10 | $\mathbf{0.2854} \pm 0.0016$ | $\mathbf{0.0114} \pm 0.0001$ | $0.9660 \pm 0.0119$ | $\mathbf{0.3084} \pm 0.0029$ | $\mathbf{0.0124} \pm 0.0001$ | $0.9775 \pm 0.0011$ |
| | | 200 | $0.3527 \pm 0.0027$ | $0.0141 \pm 0.0001$ | $0.9557 \pm 0.0165$ | $0.3824 \pm 0.0038$ | $0.0154 \pm 0.0002$ | $0.9723^* \pm 0.0039$ |
| Ours | FiLM | 10 | $0.3252 \pm 0.0020$ | $0.0130 \pm 0.0001$ | $0.9861 \pm 0.0008$ | $0.3469 \pm 0.0029$ | $0.0139 \pm 0.0001$ | $0.9862 \pm 0.0007$ |
| | Spatial | 10 | $0.3234 \pm 0.0017$ | $0.0129 \pm 0.0001$ | $0.9846 \pm 0.0022$ | $0.3489 \pm 0.0030$ | $0.0140 \pm 0.0001$ | $0.9849 \pm 0.0011$ |
| | Uncond. | 10 | $0.3181 \pm 0.0021$ | $0.0127 \pm 0.0001$ | $0.9803 \pm 0.0014$ | $0.3384 \pm 0.0011$ | $0.0136 \pm 0.0000$ | – |
| WDM | Uncond. | 1000 | $0.3073 \pm 0.0018$ | $0.0123 \pm 0.0001$ | $0.9456 \pm 0.0248$ | $0.3284 \pm 0.0013$ | $0.0132 \pm 0.0001$ | – |
| WDMa | Cond. | 1000 | $0.3166 \pm 0.0018$ | $0.0127 \pm 0.0001$ | $0.9431 \pm 0.0253$ | $0.3315 \pm 0.0030$ | $0.0133 \pm 0.0001$ | $0.9694^* \pm 0.0005$ |
| MD | Uncond. | 1000 | $0.3843 \pm 0.0026$ | $0.0153 \pm 0.0001$ | $0.9595 \pm 0.0289$ | $0.4072 \pm 0.0024$ | $0.0163 \pm 0.0001$ | – |
| MLDM | Cond. | 1000 | $0.3590 \pm 0.0021$ | $0.0144 \pm 0.0001$ | $0.9538 \pm 0.0259$ | $0.3733 \pm 0.0032$ | $0.0150 \pm 0.0001$ | $0.9784 \pm 0.0024$ |
| MOTFM | Uncond. | 10 | $0.3692 \pm 0.0024$ | $0.0147 \pm 0.0001$ | $0.9677 \pm 0.0105$ | $0.3926 \pm 0.0021$ | $0.0158 \pm 0.0001$ | – |
| MOTFMa | Cond. | 10 | $0.3747 \pm 0.0027$ | $0.0150 \pm 0.0001$ | $0.9529 \pm 0.0203$ | $0.3539 \pm 0.0033$ | $0.0142 \pm 0.0001$ | $0.9775 \pm 0.0028$ |
| BS | Cond. | – | $0.3454 \pm 0.0020$ | $0.0138 \pm 0.0001$ | $0.9346 \pm 0.0281$ | $0.3495 \pm 0.0035$ | $0.0140 \pm 0.0001$ | $0.9600 \pm 0.0097$ |

| Method | Type | Steps | Age 40–55 FID↓ | Age 40–55 MMD↓ | Age 40–55 MS-SSIM↓ | Age 65–80 FID↓ | Age 65–80 MMD↓ | Age 65–80 MS-SSIM↓ |
|---|---|---|---|---|---|---|---|---|
| Ours | RFM | 1 | $0.3511 \pm 0.0045$ | $0.0140 \pm 0.0002$ | $0.9966 \pm 0.0018$ | $0.2955 \pm 0.0043$ | $0.0118 \pm 0.0002$ | $0.9958 \pm 0.0019$ |
| | | 2 | $0.3409 \pm 0.0043$ | $0.0136 \pm 0.0002$ | $0.9927 \pm 0.0020$ | $0.2844 \pm 0.0045$ | $0.0113 \pm 0.0002$ | $0.9911 \pm 0.0026$ |
| | | 5 | $0.3303 \pm 0.0041$ | $0.0132 \pm 0.0002$ | $0.9838 \pm 0.0031$ | $0.2743 \pm 0.0043$ | $0.0109 \pm 0.0002$ | $0.9812 \pm 0.0037$ |
| | | 10 | $0.3169 \pm 0.0042$ | $0.0127 \pm 0.0002$ | $0.9580 \pm 0.0129$ | $0.2578 \pm 0.0040$ | $0.0103 \pm 0.0002$ | $0.9465 \pm 0.0145$ |
| | | 200 | $0.3160^* \pm 0.0044$ | $0.0127^* \pm 0.0002$ | $0.9560^* \pm 0.0138$ | $0.2560^* \pm 0.0041$ | $0.0102^* \pm 0.0002$ | $0.9433 \pm 0.0155$ |
| Ours | CFM | 1 | $0.3547 \pm 0.0039$ | $0.0142 \pm 0.0002$ | $0.9973 \pm 0.0009$ | $0.2971 \pm 0.0042$ | $0.0118 \pm 0.0002$ | $0.9956 \pm 0.0021$ |
| | | 2 | $0.3445 \pm 0.0043$ | $0.0138 \pm 0.0002$ | $0.9940 \pm 0.0010$ | $0.2851 \pm 0.0046$ | $0.0114 \pm 0.0002$ | $0.9915 \pm 0.0027$ |
| | | 5 | $0.3335 \pm 0.0044$ | $0.0133 \pm 0.0002$ | $0.9858 \pm 0.0018$ | $0.2741 \pm 0.0036$ | $0.0109 \pm 0.0002$ | $0.9827 \pm 0.0033$ |
| | | 10 | $0.3283 \pm 0.0043$ | $0.0131 \pm 0.0002$ | $0.9793 \pm 0.0030$ | $0.2692 \pm 0.0035$ | $0.0107 \pm 0.0001$ | $0.9754 \pm 0.0043$ |
| | | 200 | $0.3226 \pm 0.0041$ | $0.0129 \pm 0.0002$ | $0.9579^* \pm 0.0114$ | $0.2631 \pm 0.0038$ | $0.0105 \pm 0.0002$ | $0.9474^* \pm 0.0149$ |
| Ours | VP | 1 | $0.3524 \pm 0.0044$ | $0.0141 \pm 0.0002$ | $0.9969 \pm 0.0017$ | $0.2953 \pm 0.0038$ | $0.0118 \pm 0.0002$ | $0.9959 \pm 0.0019$ |
| | | 2 | $0.3416 \pm 0.0037$ | $0.0137 \pm 0.0002$ | $0.9937 \pm 0.0020$ | $0.2842 \pm 0.0040$ | $0.0113 \pm 0.0002$ | $0.9919 \pm 0.0024$ |
| | | 5 | $0.3320 \pm 0.0041$ | $0.0133 \pm 0.0002$ | $0.9854 \pm 0.0028$ | $0.2731 \pm 0.0037$ | $0.0109 \pm 0.0002$ | $0.9829 \pm 0.0031$ |
| | | 10 | $0.3258 \pm 0.0037$ | $0.0130 \pm 0.0002$ | $0.9788 \pm 0.0041$ | $0.2685 \pm 0.0041$ | $0.0107 \pm 0.0002$ | $0.9752 \pm 0.0041$ |
| | | 200 | $0.3184^* \pm 0.0040$ | $0.0127^* \pm 0.0002$ | $0.9584^* \pm 0.0134$ | $0.2596^* \pm 0.0036$ | $0.0104^* \pm 0.0001$ | $0.9472^* \pm 0.0151$ |
| Ours | Trigon. | 1 | $0.3484 \pm 0.0041$ | $0.0139 \pm 0.0002$ | $\mathbf{0.9281} \pm 0.0011$ | $0.2910 \pm 0.0041$ | $0.0116 \pm 0.0002$ | $\mathbf{0.9239} \pm 0.0011$ |
| | | 2 | $0.3168^* \pm 0.0041$ | $0.0127^* \pm 0.0002$ | $0.9611^* \pm 0.0015$ | $0.2599^* \pm 0.0039$ | $0.0104^* \pm 0.0002$ | $0.9584 \pm 0.0016$ |
| | | 5 | $0.3062 \pm 0.0036$ | $0.0123 \pm 0.0002$ | $0.9789 \pm 0.0023$ | $0.2473 \pm 0.0042$ | $0.0099 \pm 0.0002$ | $0.9756 \pm 0.0023$ |
| | | 10 | $\mathbf{0.3042} \pm 0.0041$ | $\mathbf{0.0122} \pm 0.0002$ | $0.9758 \pm 0.0028$ | $\mathbf{0.2451} \pm 0.0036$ | $\mathbf{0.0098} \pm 0.0001$ | $0.9715 \pm 0.0031$ |
| | | 200 | $0.3731 \pm 0.0049$ | $0.0149 \pm 0.0002$ | $0.9629^* \pm 0.0115$ | $0.3046 \pm 0.0049$ | $0.0122 \pm 0.0002$ | $0.9506^* \pm 0.0131$ |
| Ours | FiLM | 10 | $0.3455 \pm 0.0040$ | $0.0138 \pm 0.0002$ | $0.9860 \pm 0.0009$ | $0.2855 \pm 0.0049$ | $0.0114 \pm 0.0002$ | $0.9862 \pm 0.0008$ |
| | Spatial | 10 | $0.3450 \pm 0.0042$ | $0.0138 \pm 0.0002$ | $0.9861 \pm 0.0008$ | $0.2809 \pm 0.0039$ | $0.0112 \pm 0.0002$ | $0.9874 \pm 0.0007$ |
| | Uncond. | 10 | $0.3383 \pm 0.0026$ | $0.0135 \pm 0.0001$ | – | $0.2787 \pm 0.0017$ | $0.0111 \pm 0.0001$ | – |
| WDM | Uncond. | 1000 | $0.3277 \pm 0.0020$ | $0.0131 \pm 0.0001$ | – | $0.2694 \pm 0.0017$ | $0.0108 \pm 0.0001$ | – |
| WDMa | Cond. | 1000 | $0.3335 \pm 0.0044$ | $0.0134 \pm 0.0002$ | $0.9566^* \pm 0.0160$ | $0.2825 \pm 0.0039$ | $0.0113 \pm 0.0002$ | $0.9401 \pm 0.0208$ |
| MD | Uncond. | 1000 | $0.4074 \pm 0.0038$ | $0.0163 \pm 0.0002$ | – | $0.3426 \pm 0.0024$ | $0.0137 \pm 0.0001$ | – |
| MLDM | Cond. | 1000 | $0.3681 \pm 0.0043$ | $0.0147 \pm 0.0002$ | $0.9776 \pm 0.0034$ | $0.3311 \pm 0.0054$ | $0.0132 \pm 0.0002$ | $0.9478 \pm 0.0242$ |
| MOTFM | Uncond. | 10 | $0.3914 \pm 0.0027$ | $0.0157 \pm 0.0001$ | – | $0.3269 \pm 0.0021$ | $0.0131 \pm 0.0001$ | – |
| MOTFMa | Cond. | 10 | $0.4001 \pm 0.0048$ | $0.0160 \pm 0.0002$ | $0.9697 \pm 0.0068$ | $0.3685 \pm 0.0049$ | $0.0147 \pm 0.0002$ | $0.9630 \pm 0.0101$ |
| BS | Cond. | – | $0.3601 \pm 0.0055$ | $0.0144 \pm 0.0002$ | $0.9463 \pm 0.0286$ | $0.3188 \pm 0.0050$ | $0.0127 \pm 0.0002$ | $0.9382 \pm 0.0234$ |

Table 11: Comparison of synthetic sample quality metrics for RFM DB4 and Trigon. RK4 variants. The metrics are computed over 100 bootstrap resamples.

| Method | Type | Steps | Overall | | | Age 15–30 | | |
| --- | --- | --- | --- | --- | --- | --- | --- | --- |
| | | | FID ↓ | MMD ↓ | MS-SSIM ↓ | FID ↓ | MMD ↓ | MS-SSIM ↓ |
| Ours | RFM DB4 | 10 | $0.3142 \pm 0.0018$ | $0.0125 \pm 0.0001$ | $0.9663 \pm 0.0153$ | $0.3376 \pm 0.0031$ | $0.0136 \pm 0.0001$ | $0.9822 \pm 0.0015$ |
| Ours | Trigon. RK4 | 1 | $0.2974 \pm 0.0017$ | $0.0119 \pm 0.0001$ | $0.9485 \pm 0.0149$ | $0.3202 \pm 0.0029$ | $0.0129 \pm 0.0001$ | $0.9632 \pm 0.0008$ |
| | | 2 | $0.3112 \pm 0.0022$ | $0.0124 \pm 0.0001$ | $0.9576 \pm 0.0132$ | $0.3359 \pm 0.0031$ | $0.0135 \pm 0.0001$ | $0.9721 \pm 0.0009$ |
| | | 5 | $0.3862 \pm 0.0021$ | $0.0154 \pm 0.0001$ | $0.9604 \pm 0.0114$ | $0.4145 \pm 0.0031$ | $0.0166 \pm 0.0001$ | $0.9743 \pm 0.0016$ |
| | | 10 | $0.3820 \pm 0.0022$ | $0.0152 \pm 0.0001$ | $0.9579 \pm 0.0129$ | $0.4061 \pm 0.0035$ | $0.0163 \pm 0.0001$ | $0.9742 \pm 0.0028$ |
| | | 200 | $0.3774 \pm 0.0023$ | $0.0151 \pm 0.0001$ | $0.9518 \pm 0.0172$ | $0.4061 \pm 0.0040$ | $0.0163 \pm 0.0002$ | $0.9728 \pm 0.0042$ |

| Method | Type | Steps | Age 40–55 | | | Age 65–80 | | |
| --- | --- | --- | --- | --- | --- | --- | --- | --- |
| | | | FID ↓ | MMD ↓ | MS-SSIM ↓ | FID ↓ | MMD ↓ | MS-SSIM ↓ |
| Ours | RFM DB4 | 10 | $0.3338 \pm 0.0043$ | $0.0134 \pm 0.0002$ | $0.9804 \pm 0.0027$ | $0.2739 \pm 0.0039$ | $0.0109 \pm 0.0002$ | $0.9780 \pm 0.0035$ |
| Ours | Trigon. RK4 | 1 | $0.3175 \pm 0.0041$ | $0.0127 \pm 0.0002$ | $0.9642 \pm 0.0010$ | $0.2563 \pm 0.0037$ | $0.0102 \pm 0.0001$ | $0.9611 \pm 0.0018$ |
| | | 2 | $0.3307 \pm 0.0040$ | $0.0132 \pm 0.0002$ | $0.9716 \pm 0.0012$ | $0.2685 \pm 0.0040$ | $0.0107 \pm 0.0002$ | $0.9676 \pm 0.0021$ |
| | | 5 | $0.4080 \pm 0.0049$ | $0.0163 \pm 0.0002$ | $0.9717 \pm 0.0021$ | $0.3405 \pm 0.0048$ | $0.0136 \pm 0.0002$ | $0.9665 \pm 0.0029$ |
| | | 10 | $0.4003 \pm 0.0057$ | $0.0160 \pm 0.0002$ | $0.9697 \pm 0.0041$ | $0.3394 \pm 0.0054$ | $0.0136 \pm 0.0002$ | $0.9614 \pm 0.0056$ |
| | | 200 | $0.3974 \pm 0.0050$ | $0.0159 \pm 0.0002$ | $0.9627 \pm 0.0095$ | $0.3310 \pm 0.0045$ | $0.0132 \pm 0.0002$ | $0.9483 \pm 0.0129$ |

