# OpenReview forum: "FlowLet: Conditional 3D Brain MRI Synthesis using Wavelet Flow Matching"
_ICLR.cc/2026/Conference — Submitted to ICLR 2026_

### Official Review · Reviewer_Qxy4 · 2025-10-24

**Soundness:** 2
**Presentation:** 2
**Contribution:** 2
**Rating:** 4
**Confidence:** 5

**Summary:**

This paper introduces FlowLet, an innovative conditional generation framework that integrates flow matching with 3D wavelet transform technology to achieve age-conditional brain MRI synthesis. The study addresses demographic imbalances present in existing datasets.

**Strengths:**

The authors propose an innovative method that operates directly in the reversible wavelet domain, offering advantages in computational efficiency and anatomical fidelity while avoiding reconstruction artifacts common in potential space compression methods. The study demonstrates comprehensiveness, evaluating image quality metrics, region-based anatomical analysis, and downstream BAP performance—particularly focusing on underrepresented age cohorts.

**Weaknesses:**

The paper currently compares only a few baseline models and lacks a comprehensive evaluation of the latest mainstream brain MRI synthesis models from 2024 and 2025. It is recommended that the authors include comparisons with more recent representative models, such as TUMSyn, BrainMRDiff, BrainSynth, InBrainSyn, etc., to enhance the depth and breadth of the study in the revised manuscript.

**Questions:**

1.The paper positions "wavelet domain" and "flow matching" as core innovations, but does not explain how the synergy between the two contributes to the overall effectiveness. It is necessary to provide a specific explanation of the unique advantages of performing flow matching in the wavelet domain, compared to other approaches (such as in the original or latent space), and how this combination addresses issues that single techniques cannot solve.
2.Regarding the baseline comparison experiments, the paper selects WDM, MD, etc., but fails to explain why more recent 3D medical image generation models were not included. The criteria for baseline selection (such as representativeness, open-source availability, and task relevance) should be clarified, and an argument should be provided as to why these baselines are sufficient to cover the current mainstream technological approaches
3.The choice of Haar wavelet is based solely on its "simplicity," but the paper does not discuss whether its poor directional selectivity, frequency band aliasing, and other issues make it suitable for brain MRI generation. A systematic comparison with other wavelet bases (such as Daubechies and Symlets) should be conducted to demonstrate the advantages of Haar wavelet in the context of brain MRI generation tasks.
4.The paper mentions that latent diffusion models suffer from slow inference, but it does not fully explain how slow inference impacts clinical applications. MRI scans involve large volumes of data, and quick processing is needed in real-world medical scenarios. How does the inference time for FlowLet in generating synthetic data compare to existing models? Is FlowLet’s inference time sufficient to meet clinical demands? Although the paper states that FlowLet reduces computational demands, it does not clarify how these reductions manifest in actual deployment. How does FlowLet perform in resource-constrained clinical environments?
5.The concept of age modulation is introduced as a feature of FlowLet, but the paper does not provide sufficient explanation of how this modulation works. How does FlowLet ensure that the generated data accurately reflects aging processes at different stages? Which specific mechanisms in the model allow for accurate age-conditioned synthetic MRI generation? While the paper mentions that FlowLet reduces global variability and overfitting, it does not explain how the model prevents overfitting to specific age groups or anatomical features. What measures have been taken to ensure that the model does not generate unrealistic, biased, or overfitted synthetic data?
6.The paper uses three publicly available datasets for experiments but does not discuss any external validation beyond these datasets. How well does FlowLet generalize to datasets from different sources or in real-world clinical environments? It would be useful to understand how FlowLet performs on out-of-sample data.
7.Although FlowLet employs wavelet transforms to reduce computational demands, it does not explain how the resolution of the generated images affects their quality. Is there a trade-off between lowering resolution to improve computational efficiency and maintaining fine anatomical detail? A clearer discussion of how resolution impacts the utility of the generated data would be valuable.
8.The paper introduces several flow matching formulas, such as RFM, CFM, and VP, but does not provide enough information to explain why one formula might outperform others for specific tasks. What are the trade-offs of these formulas in terms of computational efficiency, image quality, and anatomical accuracy?
9.The paper currently compares only a few baseline models and lacks a comprehensive evaluation of the latest mainstream brain MRI synthesis models from 2024 and 2025. It is recommended that the authors include comparisons with more recent representative models, such as TUMSyn, BrainMRDiff, BrainSynth, InBrainSyn, etc., to enhance the depth and breadth of the study in the revised manuscript.

---

> ### Author Response · Authors · 2025-11-29
>
> (Please refer to our response to Reviewer Bkem for the full set of additional experiments, and to the revised paper for their complete integration.)
>
> We thank the reviewer for their detailed and insightful feedback. We address each point below and incorporate the requested clarifications into the revised paper.
>
> > Q1: The paper positions…
>
> Now we clarified how the wavelet domain and FM interact. 3D generative models face a trade-off between voxel-space fidelity (but high cost) and latent-space efficiency (but lossy compression). The 3D Haar transform offers a middle ground: it is lossless and reduces spatial size by 8×, providing efficiency without discarding anatomical information.
>
> FM aligns naturally with this representation. The wavelets separates MRI volumes into low-frequency structural components and high-frequency boundaries, so the learned velocity field operates on a representation where different spatial scales are already disentangled. This makes learning easier than in voxel space and avoids reconstruction artifacts from learned decoders.
>
> Empirically, this synergy is reflected ROI results. Superior DICE indicates that the model preserves both global structure (low-frequency subbands) and fine anatomical boundaries (high-frequency subbands). Importantly, this is achieved with only 10 steps.
>
> > Q2, Q9: Regarding the baseline…
>
> Our selection criteria were based on three main factors: 1) SotA performance in key architectural paradigms, 2) public availability of code to ensure fair and reproducible comparisons, and 3) direct relevance to our specific task.
>
> We evaluated each suggested model and provide our reasoning below, which highlights a crucial distinction between two generative paradigms:
> *   **De Novo Synthesis (Our Task):** Generating new samples from noise, conditioned on certain attributes (e.g., age). This is the goal of models in our study, which are essential for augmenting datasets with novel examples.
> *   **Image-to-Image Translation:** Taking an existing source MRI as input and modifying it to have new characteristics (e.g., aging an image, changing modality, super-resolution).
>
>
> * TUMSyn, InBrainSyn require an input MRI and cannot generate new subjects from noise.
> * BrainMRDiff lacks public code, preventing a fair and reproducible comparison.
> * BrainSynth, a recent VQ-VAE+Transformer method, has been added to our baselines.
>
> To strengthen comparisons further, and as described in our response to Reviewer Bkem, we built a new age-conditional diffusion baseline WDMa by adding our FiLM + cross-attention conditioning to the official code. This isolates the core generative framework. Full results appear in the updated tables.
>
> > Q3: The choice of Haar…
>
> We conducted a quantitative ablation (Appendix A.12) comparing five wavelet families. Haar achieved the lowest reconstruction error (~10⁻⁸), whereas smoother wavelets introduced larger and more diffuse errors.
>
> Furthermore, we have run an additional generative experiment training a FlowLet model with the smoother Daubechies-4 (db4). db4 worsened BAP (4.61 vs 4.01 MAE) and FID (0.316 vs 0.298). This further supports Haar as the optimal basis for FlowLet. See Sec. 4.4 and Tables 6–7.
>
>
> > Q4: The paper mentions…
>
> Efficiency is crucial for real-world clinical use. While we focused on data augmentation, we agree that inference speed and resource constraints are decisive factors for adoption.
>
> *   In clinical research scenarios, clinicians may need to generate hundreds of age-matched samples to build a robust statistical baseline. FlowLet generates in 1.57 seconds, whereas the best baseline (WDMa) requires ~70 seconds per volume. This ~45x speedup transforms a task that would take days into minutes.
> *   FlowLet also trains with <24 GB VRAM, enabling deployment on widely available GPUs, unlike diffusion baselines requiring 40 GB. Full VRAM and timing comparisons appear in Appendix Tables 4–5.

---

> > ### Author Response · Authors · 2025-11-29
> >
> > > Q5: The concept of age…
> >
> > In Appendix A.6 we further clarify the conditioning mechanism:
> > FiLM provides global modulation, while spatial cross-attention introduces localized, anatomy-specific refinements.
> >
> > We also provide two lines of evidence against overfitting:
> >
> > The strongest proof comes from its functional generalization on the BAP task. A model that overfits would generate samples containing spurious artifacts from the training set, which would degrade the performance of a BAP model. Since our synthetic data improves the BAP model's generalization to a held-out, independent test set of real images is compelling evidence that FlowLet is capturing true, generalizable biological patterns of aging.
> > Furthermore, our ROI analysis provides evidence of anatomical plausibility. Overfitting might manifest as generating anatomically inconsistent structures that happen to correlate with age in the training set. However, FlowLet performance on DICE and iMAE metrics, which compare synthetic samples against diverse, real age-matched references, confirms that it generates anatomically coherent and realistic structures. This demonstrates that the model is not merely replicating training set but has learned the underlying manifold of brain anatomy across different ages. Section 4.1 has been revised to clarify how these metrics further assess and mitigate overfitting.
> >
> > > Q6: The paper uses…
> >
> > While we did not validate on a completely new, unseen clinical cohort, we will argue in our limitations section that our dataset, combined from three major multi-site sources OpenBHB (10 sites), ADNI, OASIS-3, already represents significant data diversity in terms of scanners, sites, protocols, and populations. We will frame validation on entirely different clinical datasets as an important direction for future work.
> >
> > > Q7: Although FlowLet…
> >
> > We thank the reviewer for raising this insightful question regarding the relationship between resolution, computational efficiency, and anatomical detail. As clarified in Section 3, our approach handles resolution differently from methods that rely on learned compression. Unlike autoencoder-based latent models the wavelet transform is a deterministic and fully reversible operation. It does not discard information; it reorganizes it into multi-scale frequency channels. This allows FlowLet to operate on reduced spatial maps while guaranteeing exact reconstruction back to the original resolution.
> > To assess whether this efficiency compromises anatomical definition, we rely on our ROI evaluation. These metrics explicitly probe small and structurally complex regions, effectively measuring the model’s “effective resolution.” FlowLet’s strong performance on regional DICE and iMAE thus provides empirical evidence that our efficiency gains do not come at the cost of anatomical sharpness.
> > We agree that a systematic ablation across multiple output resolutions would provide additional insight. However, for fairness, all baselines in our study were trained and evaluated at the same target resolution, and extending every model to multiple resolutions was not feasible within the rebuttal period. As detailed in Appendix A.8 (VRAM and scaling tables), several baselines require substantially more memory and cannot be trained at higher resolutions without specialized hardware, making such cross-resolution experiments impractical in the current setting. Exploring resolution scaling for all methods is an interesting direction and one we plan to investigate in future work.
> >
> > > Q8: The paper…
> >
> > We add a concluding paragraph to the discussion of Table 1a that explicitly synthesizes the findings on the different FM formulations. The summary will state the trade-off as follows:
> > *   RFM/CFM (low curvature) offer the best balance of stability, performance, and simplicity, making them robust and reliable choices.
> > *   Trigonometric (high curvature) can achieve high fidelity but is less stable with simple first-order integrators, as seen in its performance degradation at high step counts.
> > *   VP connects to the powerful diffusion framework but offered no clear performance advantage in our setup to justify its increased complexity.
> > This will provide readers with a clear takeaway on why RFM is our recommended variant.
> > We thank the reviewer again for their comprehensive questions. We are confident that these clarifications and additions will significantly improve the paper.

---

### Official Review · Reviewer_Qqnb · 2025-10-25

**Soundness:** 3
**Presentation:** 3
**Contribution:** 2
**Rating:** 4
**Confidence:** 3

**Summary:**

The paper introduces FlowLet, a conditional framework for fast 3D brain MRI synthesis that combines Flow Matching with an invertible 3D Haar Discrete Wavelet Transform. Conditioning on age is implemented via a dual mechanism, Feature-wise Linear Modulation and spatial cross-attention, to steer age-specific morphology. The method targets the quality, diversity, and speed trilemma by learning velocities in wavelet space, then integrates an Ordinary Differential Equation for sampling. Empirically, the authors report competitive global fidelity at only 10 steps, improved regional anatomical plausibility, and, most importantly, gains on a downstream Brain Age Prediction  task for an underrepresented age cohort, relative to strong diffusion baselines.

**Strengths:**

- Combining FM with an invertible 3D wavelet domain for volumetric synthesis is a neat and well-motivated design choice, avoiding learned autoencoders while retaining multi-scale control. The dual conditioning for age is thoughtfully integrated and ablated.
 - The empirical study covers global distribution metrics, region-wise anatomical metrics, and a functional BAP readout. The comparison set includes both latent and wavelet diffusion baselines, plus a recent FM variant retrained with the same conditioning. Statistical testing and step-count ablations are reported, which strengthens claims about speed versus fidelity.
 - The paper is clearly written, the FM variants are laid out, and the architectural blocks and inference loop are explained with helpful figures.
 - Demonstrating that synthetic data improves BAP for older adults addresses a practical fairness issue in neuroimaging and supports the case for conditional synthesis to mitigate demographic imbalance.

**Weaknesses:**

- The paper sketches the VP connection and uses an analytically tractable conditional score to define a deterministic target velocity. However, the presentation could benefit from a concise, explicit derivation that maps the reverse-time SDE drift to the Probability Flow ODE velocity used for training, including the exact role of Tweedie-based conditioning and the factor differences in the score terms. This would make the mathematical bridge between diffusion and FM airtight for the reader.
 - Wavelet basis choice is argued from reconstruction error, not from generative stability: Haar minimizes round-trip MAE, yet it is piecewise constant. Learning smooth vector fields for ODE integration in a discontinuous basis may be suboptimal. A small generative ablation that swaps Haar for a smoother near-lossless basis, for example Coiflet-2 or Daubechies-4, and reports BAP, DICE, and solver stability would directly test whether slightly higher reconstruction error trades for better flow smoothness in practice.
 - The trigonometric path attains strong FID at low steps, then degrades at higher steps. This suggests that velocity approximation error interacts with curvature and the simple Euler integrator. A brief study with a higher order fixed-step solver or an adaptive integrator, together with curvature-aware regularization, would clarify whether the instability is from the path geometry, the network capacity, or the integrator.
 - The paper compares to latent diffusion and wavelet diffusion, and to a medical FM baseline. To strengthen the positioning, discuss or, if feasible, add results for recent conditional FM or rectified-flow variants tuned for fast sampling in high-resolution synthesis, and for very high-resolution latent diffusion scheduling strategies. Referencing recent overviews of Conditional Flow Matching and recent ultra-high-resolution latent diffusion schedulers would help calibrate novelty in efficiency-driven conditional synthesis [1, 2, 3].
 - The text claims that FlowLet trains in 24 GB while baselines need 48 GB. A short table with peak VRAM, wall-clock per iteration, samples per second, and total training time for RFM-10 versus MLDM and WDM would make the efficiency claim fully auditable.
 - Nonparametric testing and Bonferroni correction are strengths, however, reporting exact p-values and effect sizes for key pairwise differences, plus region-wise breakdowns for a small set of clinically salient structures, would improve rigor and interpretability for the regional analysis.

References:

[1] Gagneux A, Martin ST, Emonet R, Bertrand Q, Massias M. A visual dive into conditional flow matching. InThe Fourth Blogpost Track at ICLR 2025 2025.

[2] Zhang J, Huang Q, Liu J, Guo X, Huang D. Diffusion-4k: Ultra-high-resolution image synthesis with latent diffusion models. InProceedings of the Computer Vision and Pattern Recognition Conference 2025 (pp. 23464-23473).

[3] Yazdani M, Medghalchi Y, Ashrafian P, Hacihaliloglu I, Shahriari D. Flow matching for medical image synthesis: Bridging the gap between speed and quality. InInternational Conference on Medical Image Computing and Computer-Assisted Intervention 2025 Sep 20 (pp. 216-226). Cham: Springer Nature Switzerland.

**Questions:**

- Could the authors provide a compact derivation that starts from the VP reverse-time SDE, writes the corresponding Probability Flow ODE, then shows the exact conditional velocity that is implemented for training, with assumptions made explicit and the conditioning distribution spelled out?
 - What breaks first in the trigonometric variant at high step counts, the integrator or the velocity approximation, and does a second-order or adaptive solver recover stability without losing speed at 10 to 20 steps?
 - How does FlowLet scale with resolution and volume size, for example 128³ or higher? A short scaling plot for FID, DICE, and sampling time would be informative.
 - Can the authors report peak VRAM and tokens per second style throughput for training and sampling, side-by-side with MLDM and WDM, to substantiate the efficiency statement?
 - Would adding weak anatomical priors at training time, for example mask-based channels or morphometric covariates already computed for evaluation, further improve regional DICE and BAP without harming diversity?

---

> ### Author Response · Authors · 2025-11-29
>
> We thank the reviewer for their thorough and constructive feedback.
> > W1, Q1: VP connection…
>
> We improved Appendix A.2 by explicitly introducing the reverse-time VP SDE and clarifying how Flow Matching uses the deterministic probability-flow ODE. We added the missing first step of the derivation: (i) the reverse-time VP SDE, (ii) the probability-flow identity dx/dt = f - 1/2 g g^T log p_t, and (iii) its application to VP. The remaining steps (Tweedie conditional score, conditional velocity) were already present, and the appendix now provides a complete, self-contained derivation.
>
> > W2: Wavelet basis choice…
>
> To test whether smoother wavelets improve generative stability, we performed a new generative ablation study. We trained a FlowLet-RFM model using the Daubechies-4 (db4) basis and generated samples using 10 steps.
> Db4 produced worse or comparable results to Haar: FID 0.3160 vs. 0.2981, BAP MAE 4.61 vs. 4.01, and slightly lower DICE (0.4265 vs. 0.420). Haar’s sharper boundaries and lower reconstruction error yield better stability. Full metrics in Tables 1–2 and the Appendix.
>
> > W3,Q2: The trigonometric path…
>
> We tested whether instability at high step counts is due to Euler by evaluating a 4th-order Runge–Kutta solver. Results (Sec. 4.4) suggest that the challenge lies within the learned velocity field geometry rather than the solver. RK4 provided reasonable generation at very low steps, it did not yield the expected improvements at higher step counts. In fact, the RK4 produced a worse FID of 0.3819 at 10 steps compared to the 0.2854 achieved by Euler, and degraded similarly at 200 steps. The BAP performance for the RK4 model remained inferior to our primary RFM baseline (MAE 4.26 vs 4.01). Crucially, this degraded performance comes at a significant computational cost: RK4 requires 4 function evaluations per step, quadrupling the inference time compared to Euler. These findings support our choice of the low-curvature RFM trajectory. The trade-offs observed here, motivated the necessity of our extended evaluation protocol beyond standard global scores, as detailed in our response to Reviewer Bkem.
>
> > W4: The paper compares…
>
> We added [1] and [2] to Related Work to better position FlowLet with respect to recent conditional FM and high-resolution diffusion models. We note that [3] form the MOTFM baselines already included.
>
> > W5, Q4: Can the authors report peak VRAM…
>
> To make our efficiency claims concrete, we added a new table (Appendix Table 4) comparing VRAM, training and inference time across all models, enabling direct auditability of FlowLet’s efficiency.
>
> > Q3: How does FlowLet scale…
>
> All models in our comparison were trained at the same resolution (91×109×91, padded to 112³) to ensure fairness and compatibility with the BAP framework. Several baselines saturate a 48GB A6000 GPU at this resolution, so retraining all methods at higher resolutions was not feasible within the rebuttal period.
> To still address the reviewer’s question, we evaluated FlowLet’s memory footprint when increasing the input size and show them in Appendix Table 5. These measurements show that FlowLet scales favourably: it trains up to 256³ on the same GPU and maintains stable inference times (1.6s → 2.1s → 6.8s at 112³/128³/256³ for 10 steps).
>
> > Q5: Would adding weak anatomical priors…
>
> Incorporating segmentation masks or morphometric covariates is a promising direction. However, we chose not to include these priors for two task–specific reasons:
>
> *   Our primary goal is to model how morphology varies as a function of age, using age as the only conditioning variable. Providing anatomical masks would supply the model with explicit structural boundaries, making it rely on externally provided geometry rather than learning age-related morphological variation directly from the data.
> *   Using fixed masks would constrain the anatomical manifold available to the generator and reduce structural diversity. FlowLet is designed to preserve realistic variability within age, a property reflected in its DICE and BAP performance.
>
> We agree that weak priors could enhance fine-grained regional accuracy in settings where structural variability is not the primary objective. We consider this as an interesting direction for future work.
>
> > W6: Nonparametric testing…
>
> We report complete p-values in Appendix A.13.
>
> While a detailed ROI breakdown is valuable, performing this comparison (across the 95 ROIs of 3000 samples for each of the 11 models) requires recomputing segmentation outputs and rerunning the statistical pipeline for each ROI, which was not feasible during the rebuttal period. We therefore retain the aggregated regional metrics reported in Table 2b, which were computed on the full set of regions and provide a stable summary of structural fidelity across the brain.
> We view targeted analyses as a natural extension for subsequent studies and thank the reviewer for highlighting this relevant direction.

---

> > ### Author Response · Authors · 2025-11-29
> > **New additional experiments performed during rebuttal time:**
> >
> > ### Table FlowLet Scaling with Resolution
> >
> > | Resolution | Peak VRAM (BS=1) | Peak VRAM (BS=4) |
> > | :--- | :--- | :--- |
> > | 112³ | 18 GB | 22 GB |
> > | 128³ | 23 GB | 31 GB |
> > | 256³ | 42 GB | — |
> >
> > ***
> >
> > ### Table Resource Comparison
> >
> > | Resources | VRAM | Training Time (200 epochs) | Inference (3000 samples) |
> > | :--- | :--- | :--- | :--- |
> > | **FlowLet** | Batch Size (BS)4: 22GB | 80h | 1h18min |
> > | **MLDM** | Stage 1 AEKL BS2: 39GB Stage 2 Diffusion BS2: 14GB | 120h 70h (350 epochs) | 10h |
> > | **MOTFM** | BS1: 25GB | 315h | 1h40min |
> > | **MOTFMa** | BS1: 25GB | 315h | 1h40min |
> > | **Medical Diffusion** | stage 1 BS4: 16 GB stage 2 BS2: 39 GB stage 3 BS4: 10 GB stage 4 BS16: 11 GB | stage 1: 74h stage 2: 200h stage 3: 1h stage 4: 16h40min | 13h40m |
> > | **WDM** | BS1: 40GB | 96h | 58h20m |
> > | **WDMa** | BS1: 43GB | 101h | 58h20m |
> > | **BrainSynth** | stage 1 BS16: 7GB stage 2 extraction: 4GB stage 3 transformer BS2: 5GB stage 4 inference | stage 1: 205h (500 epochs) stage 2 tot: 1h stage 3: 27h (500 epochs) | 4h18m |
> >
> > ***
> >
> > ### Table ROI Metrics
> >
> > | ROI | iMAE ↓ | KLD ↓ | DICE ↑ |
> > | :--- | :--- | :--- | :--- |
> > | **Trigon. Runge Kutta 4 steps 1** | 40.1402 ± 10.5253 | 1.1861 ± 0.6764 | 0.0375 ± 0.0985 |
> > | **Trigon. Runge Kutta 4 steps 2** | 37.2138 ± 9.9648 | 0.9152 ± 0.6677 | 0.4010 ± 0.1824 |
> > | **Trigon. Runge Kutta 4 steps 5** | 37.4273 ± 10.2679 | 0.8680 ± 0.6275 | 0.4071 ± 0.1739 |
> > | **Trigon. Runge Kutta 4 steps 10** | 38.6306 ± 10.5030 | 0.9694 ± 0.7068 | 0.3697 ± 0.1659 |
> > | **Trigon. Runge Kutta 4 steps 200** | 43.3496 ± 12.0598 | 1.6140 ± 1.2770 | 0.3938 ± 0.1717 |
> > | **RFM Wavelet DB4** | 40.4824 ± 10.8453 | 1.0646 ± 0.7514 | 0.4265 ± 0.1712 |
> > | **WDMa** | 56.4303 ± 14.4431 | 2.1122 ± 1.0899 | 0.3827 ± 0.1635 |
> > | **BrainSynth** | 43.9041 ± 11.5257 | 0.8616 ± 0.5886 | 0.3559 ± 0.1583 |
> >
> > ***
> >
> > ### Table BAP
> >
> > | BAP | Train AE ↓ | Test AE ↓ |
> > | :--- | :--- | :--- |
> > | **Trigon. Runge Kutta 4 steps 1** | 1.09 ± 0.45 | 4.26 ± 3.40 |
> > | **Trigon. Runge Kutta 4 steps 2** | 0.33 ± 0.40 | 4.10 ± 3.17 |
> > | **Trigon. Runge Kutta 4 steps 5** | 0.93 ± 0.56 | 4.12 ± 3.73 |
> > | **Trigon. Runge Kutta 4 steps 10** | 0.76 ± 0.41 | 4.27 ± 3.49 |
> > | **Trigon. Runge Kutta 4 steps 200** | 0.63 ± 0.42 | 4.43 ± 3.83 |
> > | **RFM Wavelet DB4** | 0.50 ± 0.35 | 4.61 ± 4.33 |
> > | **WDMa** | 0.33 ± 0.42 | 4.93 ± 4.09 |
> > | **Brainsynth** | 0.90 ± 0.40 | 4.16 ± 3.38 |
> >
> > ***
> >
> > ### Table Global Metrics
> >
> > | Model | FID| MMD| MS-SSIM | FID (15-30) | MMD (15-30) | MS-SSIM (15-30) | FID (40-55) | MMD (40-55) | MS-SSIM (40-55) | FID (65-80) | MMD (65-80) | MS-SSIM (65-80) |
> > | :--- | :--- | :--- | :--- | :--- | :--- | :--- | :--- | :--- | :--- | :--- | :--- | :--- |
> > | **FlowLet RFM DB4 10 steps** | 0.314154 ± 0.001808 | 0.012536 ± 7.7e-05 | 0.966304 ± 0.015263 | 0.337583 ± 0.003137 | 0.013574 ± 0.000128 | 0.982178 ± 0.001504 | 0.333765 ± 0.004336 | 0.013359 ± 0.000179 | 0.980401 ± 0.002737 | 0.273923 ± 0.003937 | 0.010932 ± 0.000161 | 0.977975 ± 0.003506 |
> > | **TrigRK 4 Steps 1** | 0.297437 ± 0.001745 | 0.011868 ± 7.3e-05 | 0.948455 ± 0.014878 | 0.320217 ± 0.002903 | 0.012881 ± 0.000120 | 0.963236 ± 0.000820 | 0.317493 ± 0.004051 | 0.012708 ± 0.000168 | 0.964248 ± 0.001007 | 0.256329 ± 0.003657 | 0.010229 ± 0.000148 | 0.961069 ± 0.001781 |
> > | **TrigRK 4 Steps 2** | 0.311225 ± 0.002164 | 0.012420 ± 9.3e-05 | 0.957610 ± 0.013165 | 0.335871 ± 0.003090 | 0.013506 ± 0.000127 | 0.972096 ± 0.000870 | 0.330713 ± 0.003993 | 0.013232 ± 0.000165 | 0.971572 ± 0.001190 | 0.268544 ± 0.003963 | 0.010715 ± 0.000159 | 0.967609 ± 0.002075 |
> > | **TrigRK 4 Steps 5** | 0.386170 ± 0.002060 | 0.015400 ± 8.5e-05 | 0.960350 ± 0.011353 | 0.414506 ± 0.003143 | 0.016630 ± 0.000127 | 0.974299 ± 0.001589 | 0.408011 ± 0.004927 | 0.016311 ± 0.000202 | 0.971710 ± 0.002061 | 0.340502 ± 0.004775 | 0.013587 ± 0.000193 | 0.966490 ± 0.002860 |
> > | **TrigRK 4 Steps 10** | 0.381981 ± 0.002200 | 0.015247 ± 9.3e-05 | 0.957945 ± 0.012920 | 0.406055 ± 0.003476 | 0.016304 ± 0.000140 | 0.974157 ± 0.002764 | 0.400251 ± 0.005714 | 0.016016 ± 0.000231 | 0.969707 ± 0.004136 | 0.339380 ± 0.005372 | 0.013560 ± 0.000215 | 0.961380 ± 0.005580 |
> > | **TrigRK 4 Steps 200** | 0.377386 ± 0.002308 | 0.015068 ± 9.7e-05 | 0.951762 ± 0.017159 | 0.406057 ± 0.004026 | 0.016304 ± 0.000162 | 0.972802 ± 0.004204 | 0.397384 ± 0.005034 | 0.015902 ± 0.000205 | 0.962707 ± 0.009465 | 0.330983 ± 0.004458 | 0.013233 ± 0.000183 | 0.948324 ± 0.012871 |
> > | **WDMa** | 0.316555 ± 0.001751 | 0.012651 ± 0.000073 | 0.943097 ± 0.025341 | 0.331549 ± 0.003009 | 0.013334 ± 0.000124 | 0.969396 ± 0.005054 | 0.333507 ± 0.004434 | 0.013365 ± 0.000182 | 0.956598 ± 0.015987 | 0.282529 ± 0.003907 | 0.011298 ± 0.000158 | 0.940130 ± 0.020768 |
> > | **BrainSynth** | 0.345353 ± 0.002007 | 0.013787 ± 0.000083 | 0.934607 ± 0.028079 | 0.349529 ± 0.003455 | 0.014029 ± 0.000139 | 0.960049 ± 0.009749 | 0.360117 ± 0.005482 | 0.014418 ± 0.000222 | 0.946301 ± 0.028642 | 0.318805 ± 0.005041 | 0.012742 ± 0.000201 | 0.938182 ± 0.023414 |

---

### Official Review · Reviewer_Bkem · 2025-10-27

**Soundness:** 3
**Presentation:** 3
**Contribution:** 2
**Rating:** 4
**Confidence:** 4

**Summary:**

The presented work proposes to combine wavelet transforms and flow matching to efficiently synthesize volumetric brain MR images. Specifically, the entire flow matching generative process is conducted on images that have been encoded via Haar wavelet functions, reducing the size of all feature maps by a factor of eight. The proposed method is compared to four relevant baselines, including diffusion models incorporating wavelet transforms as well as flow matching algorithms. In addition to classical image fidelity and diversity metrics, the authors introduce three different region-based anatomical plausibility metrics that aim to measure an image's fine-grained anatomical detail. Finally, the authors evaluate the utility of generated images to train a brain age classification model. The proposed method is shown to perform on par or slightly better than the baselines while requiring fewer sampling steps.

**Strengths:**

- The combination of wavelet transforms and flow matching for efficient image synthesis is somewhat interesting and appears to be novel.

- The authors provide many details about their experimental setup. Beyond the description of the main results, they conduct extensive ablation experiments, which investigate the effect of many hyperparameters, including the used wavelet function, specific flow matching implementation, number of sampling steps, and conditioning mechanism.

- The manuscript is clearly structured, illustrated, and well written, making it easy to understand and follow.

**Weaknesses:**

- The authors motivate their work by stating that flow matching in wavelet domain increases computational efficiency. However, I do not believe that this of high importance if the ultimate goal is to generate training data for brain age prediction. Most likely, the training dataset would have to be generated only once, making slower model inference essentially irrelevant.

- Despite some concerns regarding the fairness of the experimental setup (see questions below), the proposed method only barely outperforms some of the included baselines, in particular the wavelet diffusion model.

 - I do not believe that the newly introduced region-based anatomical plausibility metrics are suited to evaluate generative models (see questions below for specific concerns).

**Questions:**

- Previous work has already combined wavelet filters with normalizing flows (Yu, Jason J., Konstantinos G. Derpanis, and Marcus A. Brubaker. "Wavelet flow: Fast training of high resolution normalizing flows." Advances in Neural Information Processing Systems 33 (2020): 6184-6196). I believe the authors should briefly discuss this work considering the conceptual similarities between normalizing flows and flow matching.

- Conditioning has long been known to improve the performance of generative models (e.g. Ho, Jonathan, and Tim Salimans. "Classifier-free diffusion guidance." arXiv preprint arXiv:2207.12598 (2022)). As such, it is noteworthy that only two of the baselines include conditioning. In order to ensure a fair comparison with the proposed method, I believe that the remaining baselines – in particular the wavelet diffusion model – should be enhanced with a conditioning mechanism similar to the authors’ MOTFMa variant of the MOTFM baseline.

- I struggle to understand the rationale behind the proposed region-based anatomical plausibility metrics. In my understanding a good generative model should yield diverse samples even for a fixed age conditioning, reflecting the anatomical variance across humans. As such, I would not expect anatomical structures to perfectly align and appear with similar intensities across samples. However, the proposed iMAE and DICE metric measure may penalize this behavior by rewarding high similarity between generated and an arbitrarily selected age-matched reference sample. Specifically, a generative model, which experiences severe mode collapse and only produces a single sample, may outperform a well-behaved model, dependent on the selected reference sample.

- On a lesser note, I presume the newly introduced anatomical plausibility metrics are reported and calculated as average across all 95 regions of interest. However, the associated variables $r$ and $\mathcal{R}$ are not properly introduced.

---

> ### Author Response · Authors · 2025-11-29
>
> We appreciate the positive feedback on novelty, experimental detail, and clarity.
>
> > W1: The authors motivate…
>
> While a dataset might be generated once for a specific task, fast synthesis is crucial in research workflows.
>
> *   Generative modeling typically involves testing many variants, conditioning strategies (age, disease, sex), and hyperparameters. A model that is ~45× faster (FlowLet: 1.57s for 10 steps vs. WDM: ~70s for 1000 steps) reduces experimentation cycles from days to hours.
> *   At scale, a 45× speedup materially reduces computational cost and energy use, aligning with Green AI principles.
>
> To make this measurable, we include in Appendix A.8 a table comparing all models on VRAM, training time, and inference time.
>
> > W2, Q2: Conditioning has long been…
>
> We re-contextualize our results to highlight that FlowLet achieves its better performance while being more efficient. This directly faces the "generative trilemma": FlowLet successfully improves sampling speed without sacrificing sample quality or diversity.
>
> The reviewer rightly points out the scarcity of conditional models in our baseline set, which reflects a general gap in the availability of such public frameworks. To address this and to provide the most rigorous comparison possible, we added two new key experiments:
>     *   We took the strongest unconditional baseline, Wavelet Diffusion Model (WDM), and extended it to be age-conditional. We call it WDMa. To achieve this, we integrated our proposed conditioning mechanism, used in FlowLet, directly into the WDM original backbone. This approach ensures a fair comparison, isolating the performance difference between the core generative frameworks and different architectures, while keeping the wavelet representation and conditioning mechanism identical.
>     *   Additionally, we also evaluated the conditional baseline BrainSynth (BS) as suggested by Reviewer Qxy4.
>
> Updated tables show that FlowLet significantly outperforms both WDMa and BS in BAP (MAE 4.01 vs. 4.93 and 4.16) and anatomical fidelity (DICE 0.426 vs. 0.382 and 0.355).
> All updated metrics appear in Tables 1a/1b, complete deatils in the Appendix.
>
> > W3, Q3, Q4: I struggle to understand the rationale…
>
> We clarify our methodology and interpretation in the revised Section 4.1.
>
> *   Standard global metrics (FID and MS-SSIM) are often insufficient for 3D MRI since the vast majority of the voxel space is empty background. This causes statistical artifacts: MS-SSIM scores are artificially inflated (>0.95) and FID scores are compressed to very low values (~0.3) simply because the background is perfectly matched across all models.
>
> Crucially, we note that most related works in the field rely exclusively on these potentially misleading global metrics. Our evaluation purposefully adds functional (BAP) and region-level (ROI) metrics to capture clinically meaningful structure.
>
> *   The reviewer’s concern about a mode-collapsed model is valid in a one-to-many setting. In our case this issue is mitigated by the evaluation protocol. Each synthetic sample is generated independently and is compared to a distinct, randomly selected, age-matched real subject. The final score is the mean over all one-to-one pairs. Under this setup a model that collapses to a single anatomical configuration would be penalized, because its unique output would need to match the anatomical variability of many different real individuals in order to obtain a favourable average score. A model that preserves inter-subject diversity, instead, produces independent samples that align with the natural variability across subjects and therefore achieves substantially better performance under the same evaluation.
> *   More importantly, we state that these anatomical fidelity metrics must be interpreted in conjunction with diversity metric (MS-SSIM). A good model must perform well on both fronts: high anatomical fidelity (low iMAE, high DICE) and high diversity (low MS-SSIM). A mode-collapsed model would exhibit high MS-SSIM (low diversity) and would be identified as a poor model, regardless of its DICE/iMAE score against a single reference.  We believe this approach, which directly probes for both anatomical and functional plausibility, offers a more complete and clinically relevant picture of a model's generative capabilities.
> *  We introduce r (individual ROI) and R (all 95 ROIs) directly in Section 4.1 for clarity.
>
> > Q1: Previous work has…
>
> We add Yu et al., “Wavelet Flow” to Related Work and clarify differences.
> Wavelet Flow uses invertible architectures and exact likelihoods. FlowLet relies on Flow Matching, which learns an ODE via regression, enabling more flexible architectures (e.g., U-Net) and more stable training in high dimensions[Flow Matching for Generative Modeling, Y. Lipman et al., ICLR 2023].
>
> We believe these additions, especially the new WDMa baseline and clarified metrics, address the reviewer’s concerns and further strengthen the paper.

---

> > ### Author Response · Authors · 2025-11-29
> > **New additional experiments (training and inference) performed during rebuttal time**
> >
> > ### Table FlowLet Scaling with Resolution
> >
> > | Resolution | Peak VRAM (BS=1) | Peak VRAM (BS=4) |
> > | :--- | :--- | :--- |
> > | 112³ | 18 GB | 22 GB |
> > | 128³ | 23 GB | 31 GB |
> > | 256³ | 42 GB | — |
> >
> > ***
> >
> > ### Table Resource Comparison
> >
> > | Resources | VRAM | Training Time (200 epochs) | Inference (3000 samples) |
> > | :--- | :--- | :--- | :--- |
> > | **FlowLet** | Batch Size (BS)4: 22GB | 80h | 1h18min |
> > | **MLDM** | Stage 1 AEKL BS2: 39GB Stage 2 Diffusion BS2: 14GB | 120h 70h (350 epochs) | 10h |
> > | **MOTFM** | BS1: 25GB | 315h | 1h40min |
> > | **MOTFMa** | BS1: 25GB | 315h | 1h40min |
> > | **Medical Diffusion** | stage 1 BS4: 16 GB stage 2 BS2: 39 GB stage 3 BS4: 10 GB stage 4 BS16: 11 GB | stage 1: 74h stage 2: 200h stage 3: 1h stage 4: 16h40min | 13h40m |
> > | **WDM** | BS1: 40GB | 96h | 58h20m |
> > | **WDMa** | BS1: 43GB | 101h | 58h20m |
> > | **BrainSynth** | stage 1 BS16: 7GB stage 2 extraction: 4GB stage 3 transformer BS2: 5GB stage 4 inference | stage 1: 205h (500 epochs) stage 2 tot: 1h stage 3: 27h (500 epochs) | 4h18m |
> >
> > ***
> >
> > ### Table ROI Metrics
> >
> > | ROI | iMAE ↓ | KLD ↓ | DICE ↑ |
> > | :--- | :--- | :--- | :--- |
> > | **Trigon. Runge Kutta 4 steps 1** | 40.1402 ± 10.5253 | 1.1861 ± 0.6764 | 0.0375 ± 0.0985 |
> > | **Trigon. Runge Kutta 4 steps 2** | 37.2138 ± 9.9648 | 0.9152 ± 0.6677 | 0.4010 ± 0.1824 |
> > | **Trigon. Runge Kutta 4 steps 5** | 37.4273 ± 10.2679 | 0.8680 ± 0.6275 | 0.4071 ± 0.1739 |
> > | **Trigon. Runge Kutta 4 steps 10** | 38.6306 ± 10.5030 | 0.9694 ± 0.7068 | 0.3697 ± 0.1659 |
> > | **Trigon. Runge Kutta 4 steps 200** | 43.3496 ± 12.0598 | 1.6140 ± 1.2770 | 0.3938 ± 0.1717 |
> > | **RFM Wavelet DB4** | 40.4824 ± 10.8453 | 1.0646 ± 0.7514 | 0.4265 ± 0.1712 |
> > | **WDMa** | 56.4303 ± 14.4431 | 2.1122 ± 1.0899 | 0.3827 ± 0.1635 |
> > | **BrainSynth** | 43.9041 ± 11.5257 | 0.8616 ± 0.5886 | 0.3559 ± 0.1583 |
> >
> > ***
> >
> > ### Table BAP
> >
> > | BAP | Train AE ↓ | Test AE ↓ |
> > | :--- | :--- | :--- |
> > | **Trigon. Runge Kutta 4 steps 1** | 1.09 ± 0.45 | 4.26 ± 3.40 |
> > | **Trigon. Runge Kutta 4 steps 2** | 0.33 ± 0.40 | 4.10 ± 3.17 |
> > | **Trigon. Runge Kutta 4 steps 5** | 0.93 ± 0.56 | 4.12 ± 3.73 |
> > | **Trigon. Runge Kutta 4 steps 10** | 0.76 ± 0.41 | 4.27 ± 3.49 |
> > | **Trigon. Runge Kutta 4 steps 200** | 0.63 ± 0.42 | 4.43 ± 3.83 |
> > | **RFM Wavelet DB4** | 0.50 ± 0.35 | 4.61 ± 4.33 |
> > | **WDMa** | 0.33 ± 0.42 | 4.93 ± 4.09 |
> > | **Brainsynth** | 0.90 ± 0.40 | 4.16 ± 3.38 |
> >
> > ***
> >
> > ### Table Global Metrics
> >
> > | Model | FID| MMD| MS-SSIM | FID (15-30) | MMD (15-30) | MS-SSIM (15-30) | FID (40-55) | MMD (40-55) | MS-SSIM (40-55) | FID (65-80) | MMD (65-80) | MS-SSIM (65-80) |
> > | :--- | :--- | :--- | :--- | :--- | :--- | :--- | :--- | :--- | :--- | :--- | :--- | :--- |
> > | **FlowLet RFM DB4 10 steps** | 0.314154 ± 0.001808 | 0.012536 ± 7.7e-05 | 0.966304 ± 0.015263 | 0.337583 ± 0.003137 | 0.013574 ± 0.000128 | 0.982178 ± 0.001504 | 0.333765 ± 0.004336 | 0.013359 ± 0.000179 | 0.980401 ± 0.002737 | 0.273923 ± 0.003937 | 0.010932 ± 0.000161 | 0.977975 ± 0.003506 |
> > | **TrigRK 4 Steps 1** | 0.297437 ± 0.001745 | 0.011868 ± 7.3e-05 | 0.948455 ± 0.014878 | 0.320217 ± 0.002903 | 0.012881 ± 0.000120 | 0.963236 ± 0.000820 | 0.317493 ± 0.004051 | 0.012708 ± 0.000168 | 0.964248 ± 0.001007 | 0.256329 ± 0.003657 | 0.010229 ± 0.000148 | 0.961069 ± 0.001781 |
> > | **TrigRK 4 Steps 2** | 0.311225 ± 0.002164 | 0.012420 ± 9.3e-05 | 0.957610 ± 0.013165 | 0.335871 ± 0.003090 | 0.013506 ± 0.000127 | 0.972096 ± 0.000870 | 0.330713 ± 0.003993 | 0.013232 ± 0.000165 | 0.971572 ± 0.001190 | 0.268544 ± 0.003963 | 0.010715 ± 0.000159 | 0.967609 ± 0.002075 |
> > | **TrigRK 4 Steps 5** | 0.386170 ± 0.002060 | 0.015400 ± 8.5e-05 | 0.960350 ± 0.011353 | 0.414506 ± 0.003143 | 0.016630 ± 0.000127 | 0.974299 ± 0.001589 | 0.408011 ± 0.004927 | 0.016311 ± 0.000202 | 0.971710 ± 0.002061 | 0.340502 ± 0.004775 | 0.013587 ± 0.000193 | 0.966490 ± 0.002860 |
> > | **TrigRK 4 Steps 10** | 0.381981 ± 0.002200 | 0.015247 ± 9.3e-05 | 0.957945 ± 0.012920 | 0.406055 ± 0.003476 | 0.016304 ± 0.000140 | 0.974157 ± 0.002764 | 0.400251 ± 0.005714 | 0.016016 ± 0.000231 | 0.969707 ± 0.004136 | 0.339380 ± 0.005372 | 0.013560 ± 0.000215 | 0.961380 ± 0.005580 |
> > | **TrigRK 4 Steps 200** | 0.377386 ± 0.002308 | 0.015068 ± 9.7e-05 | 0.951762 ± 0.017159 | 0.406057 ± 0.004026 | 0.016304 ± 0.000162 | 0.972802 ± 0.004204 | 0.397384 ± 0.005034 | 0.015902 ± 0.000205 | 0.962707 ± 0.009465 | 0.330983 ± 0.004458 | 0.013233 ± 0.000183 | 0.948324 ± 0.012871 |
> > | **WDMa** | 0.316555 ± 0.001751 | 0.012651 ± 0.000073 | 0.943097 ± 0.025341 | 0.331549 ± 0.003009 | 0.013334 ± 0.000124 | 0.969396 ± 0.005054 | 0.333507 ± 0.004434 | 0.013365 ± 0.000182 | 0.956598 ± 0.015987 | 0.282529 ± 0.003907 | 0.011298 ± 0.000158 | 0.940130 ± 0.020768 |
> > | **BrainSynth** | 0.345353 ± 0.002007 | 0.013787 ± 0.000083 | 0.934607 ± 0.028079 | 0.349529 ± 0.003455 | 0.014029 ± 0.000139 | 0.960049 ± 0.009749 | 0.360117 ± 0.005482 | 0.014418 ± 0.000222 | 0.946301 ± 0.028642 | 0.318805 ± 0.005041 | 0.012742 ± 0.000201 | 0.938182 ± 0.023414 |

---

### Official Review · Reviewer_Uwea · 2025-11-02

**Soundness:** 3
**Presentation:** 2
**Contribution:** 3
**Rating:** 6
**Confidence:** 4

**Summary:**

The paper introduces FlowLe for synthesizing age-conditioned 3D brain MRIs. The motivation is that existing MRI datasets exhibit demographic imbalances (e.g., skewed age distributions), which limit the fairness and generalizability of downstream models, such as Brain Age Prediction (BAP).
The problem definition addresses the limitations of current generative models, which are often slow (e.g., diffusion models) or introduce artifacts via latent space compression. The main novelty is the integration of Flow Matching (FM) with an invertible 3D Haar wavelet transform.
This approach operates in the wavelet domain to reduce computational demands and avoids learned compression, thereby mitigating associated artifacts.

**Strengths:**

The potential impact enables the fast synthesis (e.g., 10 steps vs. 1000 for diffusion baselines) of anatomically consistent, age-specific data, which improves BAP model performance for underrepresented age groups.

The writing quality contains objective errors, including typographical errors (e.g., "v_targot", "governed oise governed"), stray formatting characters, and data misalignment in tables (Table 2).

The evaluation amount is quantitative, comparing four variants of the proposed method (RFM, CFM, VP, Trigonometric) against five baselines using data from three datasets.

**Weaknesses:**

The paper states that "expert clinical evaluation is essential for diagnostic relevance," but this evaluation is not included.

The paper claims the architecture can "extend to multiple conditioning variables," such as disease status or cognitive scores, but this is not demonstrated. The claim of "avoiding reconstruction artifacts" is justified by avoiding latent compression; however, artifacts originating from the generative process itself are not separately quantified.

The authors identify other age-conditional models (Litrico et al., Pombo et al.) but explicitly exclude them as baselines, as they define their task differently (i.e., aging a specific MRI rather than synthesis from noise).

**Questions:**

Have you considered works like:
Yeganeh, Y., Farshad, A., Charisiadis, I., Hasny, M., Hartenberger, M., Ommer, B., Navab, N. and Adeli, E., 2025. Latent Drifting in Diffusion Models for Counterfactual Medical Image Synthesis. In Proceedings of the Computer Vision and Pattern Recognition Conference (pp. 7685-7695).

---

> ### Author Response · Authors · 2025-11-29
>
> We sincerely thank the reviewer for their time and for providing valuable and constructive feedback on our work. We appreciate the recognition of our method's potential impact. We have carefully considered all comments and will revise the manuscript accordingly to address the points raised.
> We acknowledge that clarity and precision are fundamental for communicating our work effectively. To address the concerns raised, we will perform a thorough proofread of the entire manuscript to correct any remaining typographical errors and improve overall readability.
>
> > W1: The paper states that "expert clinical evaluation is essential for diagnostic relevance," but this evaluation is not included.
>
> We agree that a full evaluation by expert neuroradiologists is valuable for validating the clinical utility and diagnostic relevance of synthetic medical images.
> However, conducting such a large-scale, multi-expert study is beyond the scope of this work and is infeasible within the rebuttal period. To address this, we take the following actions in our revised manuscript:
> * We explicitly state in our conclusion section that a full clinical evaluation by neuroradiologists represents a critical next step for clinical translation but is not included in the current study.
> * We emphasize in the Introduction that our approach goes beyond the standard global metrics (such as FID and MS-SSIM) typically relied upon in similar generative MRI literature. By validating our model via a downstream BAP task and rigorous region-based anatomical plausibility metrics (iMAE, DICE, KL), we provide a functional and anatomical assessment that standard metrics overlook. We frame these results as strong quantitative proxies for clinical relevance and for the preservation of biologically plausible morphological patterns.
>
> > W2: The paper claims the architecture can "extend to multiple conditioning variables,"...
>
> To clarify how this extension works, we add a specific explanation in Appendix A.6 detailing our MultiLabelEmbedding module. We explain that the architecture is already designed to accept multiple arguments (e.g., `--condition_vars "Age" "Sex" "Condition"`). In practice, the model creates separate linear embedders for each variable and sums their outputs to create a final `combined_cond_emb` vector. This unified vector is then fed to the U-Net's FiLM and Cross-Attention layers, integrating multiple conditions without architectural changes.
>
> > W3: The authors identify other age-conditional models (Litrico et al., Pombo et al.) but explicitly exclude them as baselines
>
> We excluded models such as Litrico et al. and Pombo et al. because they address a different, although related, task. Those approaches perform age progression on an existing MRI, which corresponds to an image-to-image translation setting, whereas our method synthesizes entirely new MRIs from random noise conditioned on age. This fundamental mismatch in inputs (a real MRI versus random noise) and in objectives makes any direct quantitative comparison of generative quality methodologically inappropriate.
>
> > Q1: Have you considered works like: …
>
> Thank you for bringing this interesting and relevant work to our attention. After careful review, we believe this work is an orthogonal contribution that does not address the core technical challenge of our paper for three critical reasons:
> *   **Dimensionality mismatch:** The work of Yeganeh et al. focuses exclusively on generating and manipulating 2D medical images (individual MRI slices and Chest X-rays). Our proposed approach explicitly addresses the significantly more complex and memory-intensive challenge of training and generating high-fidelity full 3D volumetric data.
> *   **Scope discrepancy:** Their method performs counterfactual manipulation of existing 2D slices. In contrast, our primary goal is de novo synthesis of full 3D volumes conditioned on a target age, which is essential for large-scale data augmentation and clinical simulation.
> *   **Reproducibility:** We also note that the source code for the proposed "Latent Drifting" method is not publicly available or referenced, precluding a direct experimental comparison or integration into our 3D framework.
>
> Therefore, while conceptually related, we add this citation in 4. Experimental Setup, Baselines section to better position our work, clarifying that it tackles a distinct and more challenging problem in volumetric synthesis.
>
> We thank the reviewer again for their detailed and helpful comments. We believe that incorporating these changes will substantially strengthen our manuscript.

---

### Meta-Review · Area_Chair_c3Mm · 2025-12-17

**Summary:**

Reviewers agreed the core idea is promising, but they were concerned about gaps in evaluation and positioning: missing clinical expert validation, initially limited/imbalanced conditional baselines and incomplete coverage of recent brain-MRI generators, and unclear justification for choices like Haar wavelets and the proposed ROI plausibility metrics (including potential bias toward low diversity).

The rebuttal managed to address fairness and clarity by adding an age-conditional WDM baseline (WDMa), adding BrainSynth, providing wavelet-basis and solver ablations (e.g., db4, RK4), and adding concrete VRAM/time scaling, but some concerns remain: broader external validation and the ultimate clinical interpretability of the ROI metrics without expert review.

**Reviewer Concerns:**

The rebuttal addressed major concerns by adding fairer conditional baselines (WDMa, BrainSynth), clarifying baseline selection, and substantiating efficiency claims with detailed VRAM and runtime scaling. It also strengthened methodological clarity through added derivations, wavelet-basis and solver ablations, and clearer interpretation of the ROI metrics alongside diversity measures.

Remaining concerns include the absence of expert clinical evaluation, limited demonstration of external generalization beyond the used datasets, and incomplete coverage of very recent 3D brain MRI generators and fine-grained region-wise analyses.

**Reviewer Scores:**

Reviewers are not excited about this paper in general.

---

### Decision · Program_Chairs · 2026-01-26

Reject